# Impaired peroxisomal import in *Drosophila* oenocytes causes cardiac dysfunction by inducing upd3 as a peroxikine

Kerui Huang[1], Ting Miao[1], Kai Chang[1], Jinoh Kim[1], Ping Kang[1], Qiuhan Jiang[1], Andrew J. Simmonds [2], Francesca Di Cara[3] & Hua Bai [1✉]

Aging is characterized by a chronic, low-grade inflammation, which is a major risk factor for cardiovascular diseases. It remains poorly understood whether pro-inflammatory factors released from non-cardiac tissues contribute to the non-autonomous regulation of age-related cardiac dysfunction. Here, we report that age-dependent induction of cytokine unpaired 3 (upd3) in *Drosophila* oenocytes (hepatocyte-like cells) is the primary non-autonomous mechanism for cardiac aging. We show that *upd3* is significantly up-regulated in aged oenocytes. Oenocyte-specific knockdown of *upd3* is sufficient to block aging-induced cardiac arrhythmia. We further show that the age-dependent induction of *upd3* is triggered by impaired peroxisomal import and elevated JNK signaling in aged oenocytes. We term hormonal factors induced by peroxisome dysfunction as peroxikines. Intriguingly, oenocyte-specific overexpression of *Pex5*, the key peroxisomal import receptor, blocks age-related upd3 induction and alleviates cardiac arrhythmicity. Thus, our studies identify an important role of hepatocyte-specific peroxisomal import in mediating non-autonomous regulation of cardiac aging.

---

[1] Department of Genetics, Development, and Cell Biology, Iowa State University, Ames, IA 50011, USA. [2] Department of Cell Biology, University of Alberta, Edmonton, AB T6G 2H7, Canada. [3] Department of Microbiology & Immunology, Dalhousie University, Halifax, NS B3H4R2, Canada. ✉email: hbai@iastate.edu

Age is a major risk factor for a wide range of human diseases[1] including cardiovascular diseases (CVD)[2]. During aging, cardiomyocytes undergo rapid remodeling with a variety of intracellular changes, such as impaired mitochondria, increased reactive oxygen species (ROS), and elevated inflammation[1]. The low-grade chronic and systemic inflammation (or inflammaging) is often associated with increased levels of circulating proinflammatory biomarkers (e.g., interleukin-6 (IL-6) and C-reactive protein), which are notable risk factors for CVD[3,4]. Short-term expression of IL-6 can protect myocytes from injury-induced apoptosis. However, prolonged production of IL-6 induces pathological hypertrophy and decreases cardiomyocyte contractility through the activation of Janus kinases-signal transducer and activator of transcription (JAK-STAT) signaling[5]. Elevated levels of circulating IL-6 are often associated with heart failure, myocardial damage, and atherosclerosis[5–7]. IL-6 can be produced not only by cardiomyocytes themselves in response to injury, but also by other neighboring tissues (e.g., endothelial cells), immune cells, and the liver[6,8]. However, the root causes of inflammaging, its impact on cardiac aging, and the primary sources of these inflammatory factors remain to be determined.

The liver is a major endocrine organ that produces a variety of systemic factors to coordinate body's physiology and metabolism. It can produce proinflammatory cytokine IL-6 upon infection or injury[9]. Patients with liver dysfunction, such as cirrhosis, often show increased cardiac arrhythmias. Furthermore, nonalcoholic fatty liver disease is a strong risk factor for cardiomyopathy[10]. About 30% of alcoholic hepatitis patients develop cardiomyopathy and organ failure. Together, these evidences suggest a potential cross-talk between liver and heart. It is known that aging significantly alters liver morphology and function[11]. Recently, using Drosophila oenocytes as a hepatocyte model, we observed a similar downregulation of oxidative phosphorylation, and upregulation of inflammatory signaling in aged fly oenocytes[12]. However, it remains unclear whether liver inflammation directly influences heart function at old ages.

The liver is known to enrich with the peroxisome, a key organelle for ROS metabolism, alpha and beta oxidation of fatty acids, biosynthesis of ether phospholipids[13]. The peroxisome assembly and the import of peroxisomal matrix proteins are controlled by a group of peroxisomal proteins called peroxins (PEXs). Mutations in PEXs disrupt normal peroxisome function and cause peroxisome biogenesis disorders, such as Zellweger syndrome[14]. Several studies suggest that peroxisomal import function declines with age[15–17]. Consistently, our recent translatomic analysis shows that the majority of peroxisome genes are downregulated in aged fly oenocytes[12]. However, the role of peroxisome in aging regulation is unclear.

Our findings here demonstrate a peroxisome-mediated interorgan communication between the oenocyte and the heart during Drosophila aging. We find that elevated ROS in aged oenocytes promotes cardiac arrhythmia by inducing unpaired 3 (upd3), an IL-6-like proinflammatory cytokine[18]. Either decreasing the expression of upd3 in oenocytes or blocking the activation of JAK-STAT signaling in cardiomyocytes alleviates aging- and oxidative stress-induced arrhythmia. Finally, we show that peroxisomal import function is disrupted in aged oenocytes. Knockdown (KD) of cargo receptor Pex5 triggers peroxisomal import stress (PIS), which induces upd3 expression through c-Jun N-terminal kinase (JNK) signaling in oenocytes. On the other hand, oenocyte-specific overexpression of Pex5 restores peroxisomal import blocks age-induced upd3 and cardiac arrhythmicity. Together, our studies reveal a nonautonomous mechanism for cardiac aging that involves in hepatic peroxisomal import-mediated inflammation.

## Results

**Oenocyte ROS homeostasis modulates cardiac function.** Disrupted ROS homeostasis is one of the hallmarks of aging[19]. Our recent translatomic analysis in Drosophila oenocytes (a hepatocyte-like tissue) revealed an overall downregulation of antioxidant genes under aging, which was consistent with elevated oxidative stress in this tissue[12]. To determine whether redox imbalance in oenocytes can nonautonomously impact cardiac function, we first induced oxidative stress specifically in oenocytes of female flies by crossing the PromE-Gal4 driver[20] to RNAi lines against ROS scavenger genes Catalase (Cat) and Superoxide dismutase 1 (Sod1) (Supplementary Fig. 1a, b). Heart contractility was then assessed using the semiautomatic optical heartbeat analysis (SOHA). By crossing to UAS-GFP lines, we showed that PromE-Gal4 driver is specifically active in oenocytes of female flies (Supplementary Fig. 1c–e). Interestingly, oenocyte-specific KD of Cat or Sod1 resulted in an increase in cardiac arrhythmicity, as measured by arrhythmia index (AI) (Fig. 1a). These results suggest that disrupted ROS homeostasis in Drosophila oenocytes can modulate cardiac rhythm through an unknown nonautonomous mechanism.

Next, we asked whether heart function could be protected from oxidative stress and aging by maintaining redox balance in oenocytes. We first induced ROS level systemically by feeding flies with paraquat (PQ), an oxidative stress inducing agent. Feeding flies with PQ for 24 h induced ROS level in oenocytes, as measured by dihydroethidium (DHE) staining (Fig. 1b, c). Consistent with the previously report[21], PQ feeding also induced arrhythmicity in fly hearts (Fig. 1d, e). Intriguingly, using an oenocyte-specific GeneSwitch driver (PromE$^{GS}$-Gal4, Supplementary Figs. 1d and 2a), overexpression of Sod1 in adult oenocytes (PromE$^{GS}$-Gal4>UAS-Sod1$^{OE}$) was sufficient to block PQ-induced ROS production in oenocytes (Fig. 1b, c), as well as alleviated PQ-induced arrhythmicity in the heart (Fig. 1d, e). Similarly, overexpressing Sod1 in oenocytes attenuated aging-induced cardiac arrhythmicity (Fig. 1g, h). RU486 (mifepristone, or RU) was used to activate PromE$^{GS}$-Gal4 driver (+RU), whereas control genotype is the same, but with no RU feeding (−RU) (Supplementary Fig. 2a). RU486 feeding alone did not significantly affect cardiac arrhythmia (Supplementary Fig. 2c–e). To examine whether Sod1-mediated cardiac protection is specific to oenocytes, we crossed Sod1 overexpression line to a fat body (FB)/gut-specific GeneSwitch driver S106$^{GS}$-Gal4[22] (Supplementary Fig. 2b). Overexpression of Sod1 in FB and gut did not rescue PQ-induced arrhythmia (Fig. 1f). Together, these data suggest that oenocytes play a specific and crucial role in maintaining cardiac health during aging and PQ-induced oxidative stress, likely through an unknown circulating factor.

**Oenocyte upd3 mediates aging- and PQ-induced arrhythmia.** To identify factors that are secreted from oenocytes and communicate to the heart to regulate cardiac function during aging and oxidative stress, we first compared the list of Drosophila secretory proteins[23] with our recent oenocyte translatomic data set[12]. We identified 266 secretory factors that are differentially expressed in aged (4-week-old) or PQ-treated oenocytes (Fig. 2a). Among these secretory factors, we selected 27 candidates that encode for cytokines and hormonal factors in a reverse genetic screen to determine their roles in mediating oenocyte–heart communication under oxidative stress. KD of several candidate factors (e.g., sala, BG642167) in oenocytes induced cardiac arrhythmia (Supplementary Fig. 3a), similar to the KD of Cat and Sod1. On the other hand, our genetic screening identified four candidates whose KD specifically in oenocytes significantly attenuated PQ-induced cardiac arrhythmicity (Fig. 2b). The four candidate genes are PGRP-SB1, Ag5r2, TotA, and upd3. We

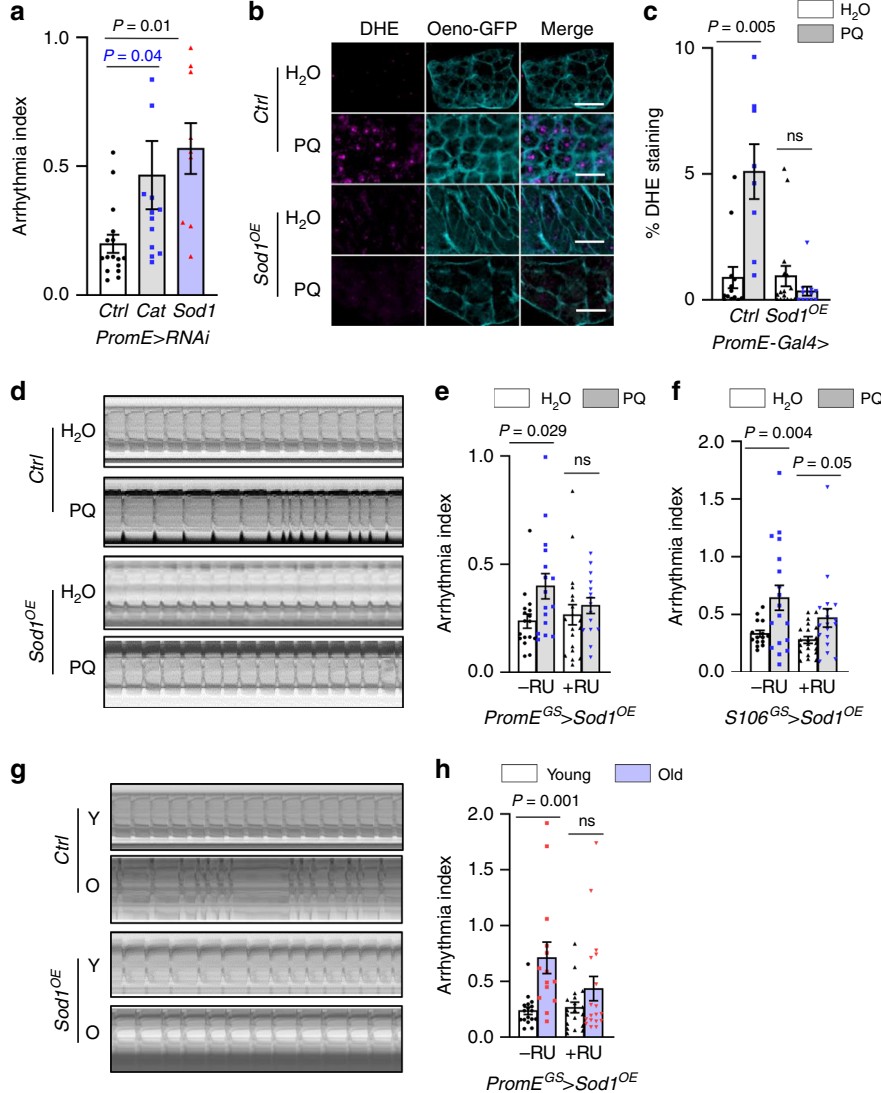

**Fig. 1 Oenocyte ROS homeostasis non-autonomously modulates cardiac function. a** Arrhythmia index of oenocyte-specific *Cat* (n = 9) and *Sod1* (n = 13) knockdown flies (1-week-old). *Ctrl* genotype is *PromE>attP40* (n = 16). **b** Representative images of ROS levels in dissected oenocytes from flies fed on normal diet (white bar) or 10mM paraquat (grey bar). All flies express mCD8::GFP under *PromE-Gal4*. *Sod1* was specifically overexpressed in the oenocytes (*Sod1*$^{OE}$). Scale bar: 20 µm. **c** Quantification of the percentage of DHE-positive staining in region of interest ROIs from 5 flies (n$_{left-right}$ = 13, 8, 12, 16 ROIs). **d** Representative M-mode showing heart contraction in control and *Sod1* overexpression flies fed on normal or 10mM paraquat food. *Sod1* was expressed using the GeneSwitch *PromE$^{GS}$-Gal4* (+RU). *Ctrl* genotype is *PromE$^{GS}$>Sod1*$^{OE}$ with no RU (−RU). **e** Arrhythmia index of control and oenocyte-specific Sod1 overexpression flies fed on normal or 10 mM paraquat diets (n$_{left-right}$ = 17, 16, 19, 15 flies). **f** Arrhythmia index of control and fat body/gut-specific Sod1 (*S106-Gal4>Sod1OE*) overexpression flies fed on normal or 10 mM paraquat diets. Overexpression specifically in fat body and gut (n$_{left-right}$ = 15, 18, 21, 17 flies). **g** Representative M-mode showing heart contraction in young (2 weeks, white bar) and old (6 weeks, purple bar) flies with or without oenocyte-specific *Sod1* overexpression. *Ctrl* genotype is *PromE$^{GS}$>Sod1*$^{OE}$ with no RU. **h** Arrhythmia index of control and oenocyte-specific *Sod1*$^{OE}$ flies at young and old ages (n$_{left-right}$ = 17, 19, 14, 18 flies). Data are represented as mean ± SEM. *P* values are calculated using either two-way ANOVA (**c**, **e**, **f**, **h**) or one-way ANOVA (**a**), followed by Holm-sidak multiple comparisons. ns: not significant.

further verified our screening results using oenocyte-specific GeneSwitch driver (*PromE$^{GS}$-Gal4*) and repeated the KD experiments for *PGRP-SB1* (Supplementary Fig. 3b) and *upd3* (Fig. 2c, two independent *upd3* RNAi lines used). The KD efficiency of *upd3* RNAi was verified by quantitative RT-PCR (QRT-PCR) (Supplementary Fig. 5a). Consistent with the screening results, KD of *PGRP-SB1* and *upd3* in adult oenocytes blocked PQ-induced arrhythmia.

Among four identified secretory factors, *upd3* is a proinflammatory factor that belongs to four-helix bundle IL-6 type cytokine family[18]. In *Drosophila*, *upd3* is one of the three ligands that activate JAK/STAT signaling pathway. The expression of

*upd3* in oenocytes was higher than two other unpaired proteins (upd1 and upd2), and *upd3* expression was significantly induced under aging (Fig. 2d). Although the expression of *upd2* was slightly induced in aged oenocytes (Fig. 2d), *upd2* KD did not block PQ-induced arrhythmicity (Fig. 2b). Furthermore, we found that *upd3* transcripts can be detected in several other adult tissues besides oenocytes, such as abdominal FB, gut, and ovary (OV) (Fig. 2e). At young ages, the highest expression of *upd3* was found in the gut (Fig. 2e). Intriguingly, *upd3* expression in oenocytes increased sharply during normal aging (more than 52-fold) and became the highest among all adult tissues at old ages. *upd3* expression did not show age-dependent increases in gut and

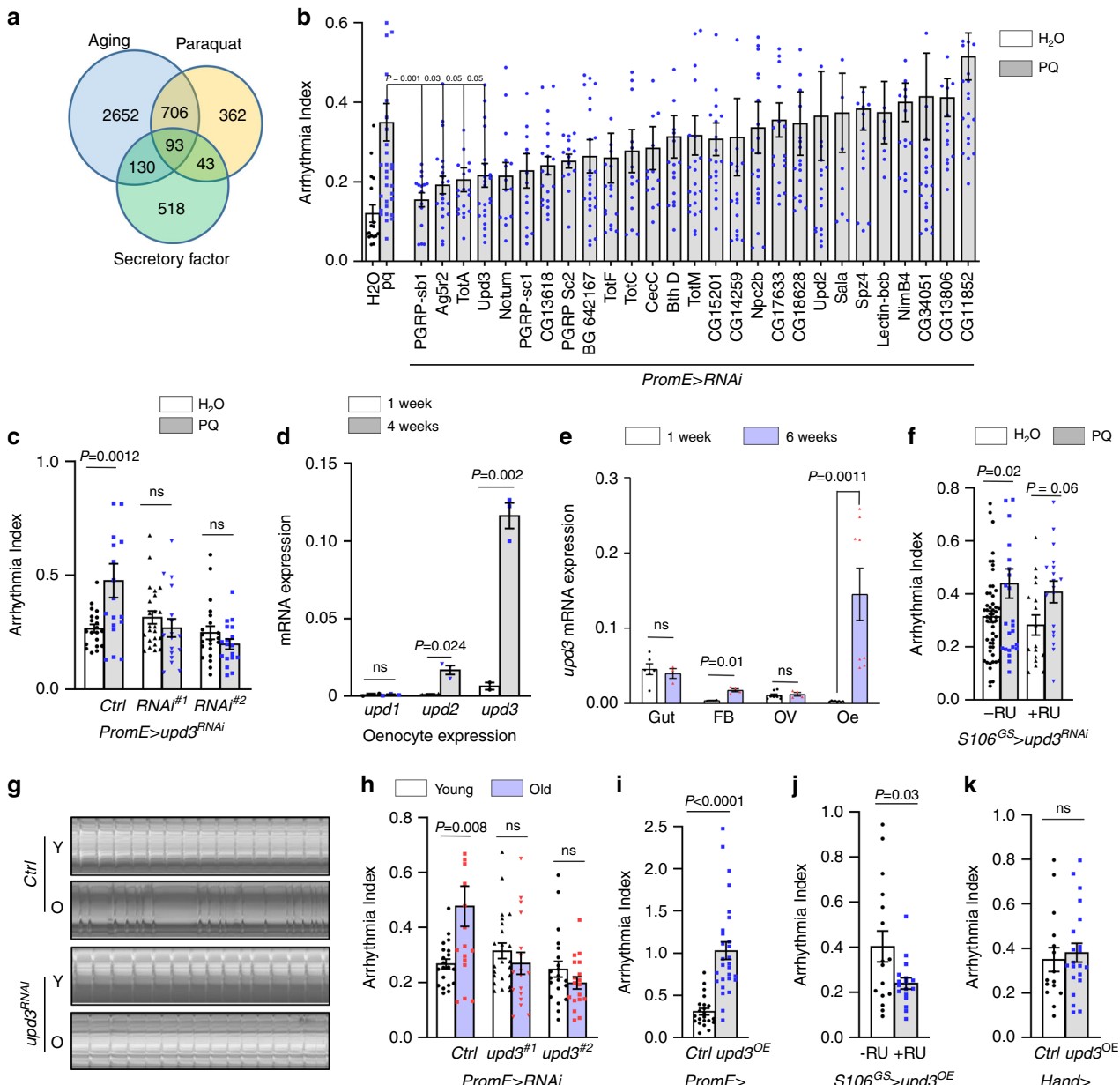

**Fig. 2 Pro-inflammatory upd3 produced from oenocytes mediates arrhythmia. a** Venn diagram showing the number of the predicted secretory proteins that are differentially expressed (≥2-fold, FDR < 0.05) under aging and paraquat treatment. Aging and paraquat RNA-Seq data were from our previous studies. Fly ages: 10-day-old vs. 30-day-old. **b** Genetic screening on 27 candidate genes for their role in paraquat-induced arrhythmia. WT: Wild-type (*attP2* or *attP40* RNAi control lines). For statistical numbers, refer to the Methods section. **c** PQ-induced arrhythmia measured by SOHA for two independent *upd3* RNAi lines under oenocyte-specific GeneSwitch driver (*PromE^GS-Gal4*). Ctrl genotype is *PromE>attP40*. (n_left-right = 20, 18, 23, 17, 22, 18 flies). **d** Relative mRNA expression of *upd1, upd2* and *upd3* from isolated oenocytes at ages of 1 week or 4 weeks. N = 3 biologically independent samples. **e** Relative mRNA expression of *upd3* in different tissues dissected from young (1 week) and old (6 weeks) female flies. FB: fat body, OV: ovary, oe: oenocytes. N = 4 biological samples, results pooled from 2 independent experiments. **f** PQ-induced arrhythmia measured by SOHA for *upd3* RNAi under fat body/gut-specific GeneSwitch driver, (n_left-right = 52, 27, 17, 18 flies). **g** Representative M-mode traces of wild-type and oenocyte-specific upd3 knockdown flies at young and old ages. **h** Arrhythmia index of wild-type and oenocyte-specific upd3 knockdown flies. Two independent RNAi lines used. Ctrl genotype is *PromE>attP40*, (n_left-right = 14, 17, 19, 20, 18, 19 flies). **i** Arrhythmia index for flies with ectopic upd3 expression (UAS-upd3-GFP) specifically in oenocytes. Flies are 1-week-old. Ctrl genotype is *PromE>attP40* (n_left-right = 22, 28 flies). **j** Arrhythmia index of flies overexpressing upd3 in fat body and gut (S106GS-Gal4), n_left-right = 16, 17 flies. **k** Arrhythmia index of flies overexpressing upd3 in the heart (Hand-Gal4), n_left-right = 16, 21 flies. Data are represented as mean ± SEM. P values are calculated using two-tailed unpaired t-test (**i–k**), one-way ANOVA (**b**), or two-way ANOVA (**c–f, h**) followed by Holm-Sidak multiple comparisons. ns: not significant.

OV, and it only slightly increased in aged FB (Fig. 2e). These findings suggest that oenocytes are the primary source of *upd3* production in aged flies.

Because *upd3* expressed in multiple adult tissues, we wonder if upd3 produced from tissues other than oenocytes also

contributes to the nonautonomous regulation of cardiac function. To test this tissue-specific effect, we knocked down *upd3* using the FB/gut-specific driver (*S106^GS-Gal4*) and found that FB/gut-specific *upd3* KD did not alleviate PQ-induced arrhythmia (Fig. 2f). Similar to the findings from PQ treatment

(Fig. 2c), oenocyte-specific *upd3* KD also blocked aging-induced cardiac arrhythmia (Fig. 2g, h, two independent *upd3* RNAi lines showed). Conversely, oenocyte-specific overexpression of *upd3* at young ages induced premature cardiac aging phenotypes (high AI) (Fig. 2i), which is similar to age-induced cardiac arrhythmia seen in multiple control flies (Supplementary Fig. 3c). In contrast, FB/gut- or cardiac-specific overexpression of *upd3* did not induce arrhythmia (Fig. 2j, k). Interestingly, FB/gut overexpression of *upd3* slightly decreased arrhythmia (Fig. 2j). Taken together, these results suggest that *upd3* is the primary cytokine that is secreted from oenocytes to regulate aging- and stress-induced cardiac arrhythmicity.

**Oenocyte upd3 induce arrhythmia through JAK-STAT pathway.** upd3 is known to systemically upregulate JAK-STAT pathway in response to tissue injuries, excess dietary lipid, PQ treatment, and bacterial infection[24–26]. We asked whether oenocyte-produced upd3 can signal to the heart and activate JAK-STAT pathway in cardiomyocytes. To test this idea, we used the nuclear localization of Stat92E (fly homology of mammalian STAT transcription factors) to indicate the activation of JAK-STAT signaling. Consistent with previous findings[26,27], PQ treatment induced the levels of transcription factor Stat92E and promoted its localization near to the nucleus in heart tissue (Fig. 3a, b). Interestingly, we found that oenocyte-specific *upd3* KD attenuated PQ-induced Stat92E in the heart (Fig. 3a, b). To further confirm the activation of cardiac JAK-STAT by oenocyte-produced upd3, we examined the transcription of *Socs36E*, a key Stat92E target gene. Consistent with the Stat92E immunostaining, the cardiac expression of *Socs36E* was induced by PQ treatment, while oenocyte-specific *upd3* KD diminished it (Fig. 3c). Age-dependent induction of *Socs36E* in the heart was also attenuated by oenocyte-specific *upd3* KD (Fig. 3d). We attempted to verify above findings using Stat93E-GFP reporters (*2XStat92E-GFP* and *10XStat92E-GFP*)[28], however PQ treatment did not significantly induce the reporter activities (Supplementary Fig. 4a–c).

Activation of JAK-STAT plays a significant role in the pathogenesis of myocardial ischemia and cardiac hypertrophy[29,30]. We then asked whether blocking JAK-STAT signaling in fly hearts could protect cardiac function under oxidative stress and aging. As expected, we found that heart-specific activation of JAK-STAT signaling by expressing an active form of JAK kinase *hopscotch/hop* (*hop^{Tuml}*) in young fly heart (using heart-specific driver *Hand-gal4*) induced cardiac arrhythmia (Fig. 3e). Conversely, cardiac-specific KD of either the receptor *domeless (dome)* or the transcription factor *Stat92E* blocked PQ-induced cardiac arrhythmia (Fig. 3f, g). Similarly, KD of *Stat92E* and *hop* in the heart attenuated aging-induced arrhythmia (Fig. 3h, i).

Next, we asked whether oenocyte-produced upd3 is secreted into the hemolymph and directly targeted cardiomyocytes. We overexpressed upd3-GFP fusion proteins specifically in the oenocytes and analyzed the hemolymph samples using western blotting. Using an anti-GFP antibody (with some nonspecific cross-reactivities, Supplementary Fig. 3d), the upd3-GFP fusion proteins were successfully detected in the hemolymph extracted from *PromE>upd3-GFP* flies (Fig. 3j, k), suggesting the oenocyte-produced upd3 indeed can be secreted into the hemolymph. Interestingly, free GFP proteins were also found in the hemolymph, which may be due to a cleavage of the C-terminus of upd3 occurring after its secretion (Fig. 3j, k). It is known that the activities of many mammalian cytokines are regulated by proteolytic processing[31]. Human and murine IL-6 is known to be cleaved by meprin metalloproteases at its c-terminus[32]. Together, these data suggest that upd3 produced from oenocytes is released into the hemolymph and activates JAK-STAT in the heart to regulate cardiac function.

**Oenocyte peroxisome import stress induces *upd3* and arrhythmia.** Next, we asked how aging upregulates *upd3* expression in oenocytes. In our previous oenocyte translatomic analysis, we found that genes involved in oxidative phosphorylation and peroxisome biogenesis are significantly downregulated during aging, which are consistent with elevated ROS levels in aged oenocytes[12]. It is known that mitochondria and peroxisomes are two major ROS contributors. We then investigated whether age-dependent downregulation of genes in mitochondrial respiratory chain complexes and peroxisome biogenesis contributes to upd3 overproduction and oenocyte–heart communication. Interestingly, oenocyte-specific KD of mitochondrial complex I core subunit *ND-75* and mitochondrial manganese superoxide dismutase *Sod2* showed no effects on cardiac arrhythmia (Fig. 4b). The KD efficiency of *ND-75* RNAi was verified by QRT-PCR (Supplementary Fig. 5b). On the other hand, KD of the key factors (peroxines) involved in peroxisomal import process (*Pex5, Pex1*, and *Pex14*) in oenocytes significantly induced cardiac arrhythmia (Fig. 4a, b). The KD efficiency of *Pex1* RNAi, *Pex5* RNAi, and *Pex14* RNAi was verified by QRT-PCR (Supplementary Fig. 5c–e). Pex5 is the key import factor that binds to cargo proteins containing peroxisomal targeting signal type 1 (PTS1) and delivers them to peroxisomal matrix through Pex13/Pex14 docking complex and Pex1/Pex6 recycling complex[33,34] (Fig. 4a). Interestingly, Pex5 itself is the major component of the peroxisomal translocon (also known as importomer, or import pore), which interacts with Pex14 and translocates cargo proteins across peroxisomal membrane through an ATP-independent process[35–37]. In addition, we noticed that not all peroxisome genes were involved in oenocyte–heart communication. Oenocyte-specific KD of *Pex19*, the key peroxisomal membrane assembly factor, did not promote cardiac arrhythmia (Fig. 4a, b). The KD efficiency of *Pex19* RNAi was verified by QRT-PCR (Supplementary Fig. 5f). Interestingly, knocking down of either *Pex5* or *Pex1*, but not *Pex19*, induced the levels of ROS in oenocytes (Supplementary Fig. 5h–j). The differential regulation on ROS metabolism by different peroxines might explain their distinct roles on cardiac arrhythmia and oenocyte–heart communication.

Our previous oenocyte translatomic analysis found that many peroxisomal matrix enzymes were significantly downregulated during aging, such as peroxiredoxin-5 (*Prx5*, a thioredoxin peroxidase regulating hydrogen peroxide ($H_2O_2$) detoxification), dihydroxyacetone phosphate acyltransferase (*Dhap-at*, the key enzyme catalyzing the first step in the biosynthesis of ether phospholipids, such as plasmalogens), alkyldihydroxyacetonephosphate synthase (*ADPS*, another key enzyme involved in ether phospholipid biosynthesis), and acyl-CoA oxidase (Acox57D-d and Acox57D-p, the rate limiting enzyme for peroxisomal beta oxidation and its deficiency has been previously linked to IL-6 induction)[38]. Interestingly, KD of *Prx5* induced cardiac arrhythmia, but not *Dhap-at, ADPS, Acox57D-d*, and *Acox57D-p* (Fig. 4c). Thus, these data suggest that peroxisome-mediated ROS homeostasis in oenocytes plays a major role in mediating oenocyte–heart communication.

We then asked whether impaired peroxisomal import in oenocytes influences *upd3* expression. As expected, KD of either *Pex5* or *Pex1* in oenocytes induced the mRNA levels of *upd3* (Fig. 4d). Similarly, KD of peroxisomal antioxidant gene *Cat* significantly upregulated *upd3* transcription, whereas KD of *Pex19* and *ND-75* did not alter *upd3* expression (Fig. 4d). Interestingly, two other unpaired proteins *upd1* and *upd2* remained unchanged under *Pex5* KD (Fig. 4e). Given that impaired peroxisomal function induces *upd3* expression in oenocytes, we speculate that *Pex5* KD in oenocytes could remotely regulate JAK-STAT activity in the heart. Indeed, we found that oenocyte-specific *Pex5* KD-induced *Stat92E* expression and Stat92E nuclear localization in the heart (Fig. 4f, g). Together, our findings showed that impaired

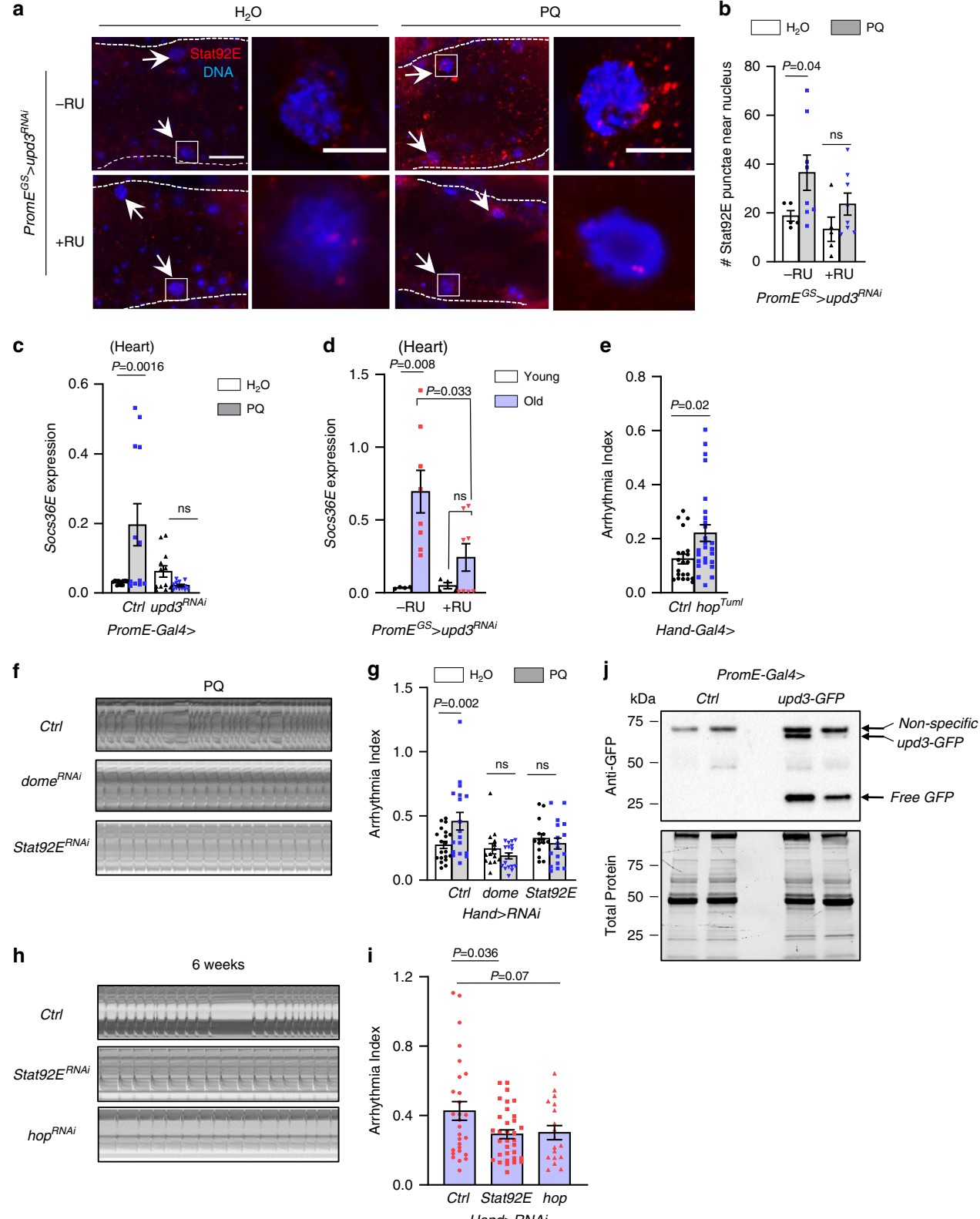

peroxisomal import promotes upd3 production in oenocytes and nonautonomously induces cardiac JAK-STAT and arrhythmia.

Lastly, we characterized peroxisome tissue distribution in adult female flies. Peroxisomes are marked by Pmp70, a peroxisomal membrane protein involved in the transport of long-chain acyl-CoA across peroxisomal membrane[39]. We noticed that

peroxisome showed tissue-specific enrichment in adult *Drosophila*. The peroxisome abundance was the lowest in the heart and highest in oenocytes, while the number of peroxisomes in FB (adipose-like) and pericardial cells (podocyte-like) was in the middle range (Fig. 4h, i). High enrichment of peroxisome is a key feature of mammalian liver[13]. Interestingly, unlike oenocyte-

**Fig. 3 Oenocyte upd3 activates JAK-STAT pathway in cardiomyocytes to induce arrhythmia. a** Representative images of Stat92E immunostaining in the heart of oenocyte-specific *upd3* KD ¬(+RU) and control flies (−RU). Flies were treated with normal or paraquat diet for 24 h. Arrows indicate cardiomyocyte nuclei. White boxes indicate the regions shown in the right insets. Scale bar: 20 μm (inset: 5 μm). **b** Quantification of the Stat92E-positive punctae near cardiomyocyte nucleus. N = 7 flies. Data presented are representative of two independent experiments. **c**, **d** QRT-PCR analysis on *Socs36E* mRNA expression in the heart of control (*attP40*) and flies with oenocyte-specific *upd3* KD under either paraquat treatment or aging. In paraquat treatment (**c**), 4 independent biological samples are shown, results are pooled from 3 experiments. In aging study (**d**), flies with two ages (1 week vs 5 weeks) are collected. N = 4 biological samples from 2 independent experiments. **e** Arrhythmia index of young flies (1-week-old) with heart-specific expression of an activated form of hop (*hop$^{Tuml}$*) ($n_{left-right}$ = 21, 26 flies). **f** Representative M-mode of paraquat-treated wild-type, heart-specific dome and *Stat92E* RNAi flies. **g** Arrhythmia Index of paraquat-treated wild-type, heart-specific dome and *Stat92E* KD flies. Ctrl genotype is *Hand>attP40* ($n_{left-right}$ = 20, 18, 14, 17, 15, 17 flies). **h** Representative M-mode of heart-specific *Stat92E* and *hop* RNAi flies at old ages. **i** Arrhythmia Index of heart-specific *Stat92E* and *hop* RNAi flies at old ages. Ctrl genotype is *Hand>attP40*. ($n_{left-right}$= 28, 33, 18 flies). **j** Western blot analysis on the hemolymph samples extracted from flies expressing upd3-GFP fusion proteins in oenocytes. Two biological replicates are shown. Total protein loaded onto the Bio-Rad Stain-Free gel was visualized using ChemiDoc MP Imagers after UV activation. Ctrl genotype is *PromE>attP40*. **k** Quantification of western blots in **j**. The data represent the intensity of GFP bands normalized to the total protein. Data are represented as mean ± SEM. *P* values are calculated using two-tailed unpaired t-test (**e**, **k**) or one-way ANOVA (**i**), or two-way ANOVA (**b**–**d**, **g**) followed by Holm-Sidak multiple comparisons. ns: not significant.

specific manipulation, knocking down of *Pex5* in FB and gut using *S106$^{GS}$-Gal4* did not induce cardiac arrhythmia (Fig. 4j). These results suggest that oenocyte-specific peroxisomal import contributes significantly to the hyperproduction of proinflammatory cytokine and nonautonomous regulation of cardiac function. Here, we refer to the cellular stress responses (e.g., production of inflammatory cytokines) caused by impaired peroxisomal import as PIS. We term PIS-induced hormonal factors as peroxikines, which are produced and released in response to PISs to modulate cellular homeostasis in distant tissues.

**Peroxisome import stress induces *upd3* through JNK pathway.** We next examined the molecular mechanism by which *Pex5* KD-mediated PIS induces the expression of peroxikine *upd3*. It is known that upd3 plays an important role in activating innate immunity and tissue repair upon infection. A recent genetic screen identified several transcription factors that are responsible for infection-induced *upd3* transcription, such as mothers against dpp (Mad), the AP-1 complex (kayak/kay and Jun-related antigen/Jra), and yorkie (yki)[40]. Interestingly, we found that oenocyte-specific KD of *Pex5* induced the expression of *kay* and *Jra*, but not *Mad* and *yki* (Fig. 5a), suggesting kay/Jra may be the transcription factors regulating *upd3* expression upon PIS. In *Drosophila*, kay and Jra form a transcription complex, similar to the AP-1 complex in mammals, to mediate JNK signaling[41,42]. We then asked whether PIS can also activate JNK signaling. Through immunostaining analysis, we found that *Pex5* KD induced the phosphorylation of JNK in oenocytes near the nucleus (Fig. 5b, c). To further confirm above findings, we performed western blots on dissected oenocytes to assess the levels of P-JNK using an anti-phospho-JNK (Thr183/Tyr185) antibody. We found that *Pex5* KD slightly increased the phosphorylation of JNK (Fig. 5d, e). On the other hand, we utilized a JNK reporter (TRE-DsRedT4) to monitor the transcription activity of AP-1 complex[43]. *Pex5* KD significantly induced the transcription activity of AP-1 (kay/Jra) in oenocytes (Fig. 5f). In addition, we confirmed this finding by measuring the mRNA expression of *puckered* (*puc*), the downstream target gene of kay. Consistently, *puc* expression was also induced by oenocyte-specific *Pex5* KD (Fig. 5g). Interestingly, the activation of JNK pathway, indicated by the upregulation of *kay* and *puc*, was also observed in flies with oenocyte-specific *Cat* KD, but not *ND-75* KD (Fig. 5h). This further suggests that peroxisome dysfunction plays a key role in activating JNK pathway. Finally, to directly examine the role of kay in mediating PIS-induced *upd3* transcription, we generated fly lines with *Pex5; kay* double KD. The KD efficiency of *kay* RNAi was verified by QRT-PCR (Supplementary Fig. 5g). As expected, KD of *kay* completely blocked *Pex5* KD-induced *upd3*

transcription in oenocytes (Fig. 5i), suggesting that *Pex5* KD-mediated PIS upregulates peroxikine *upd3* through JNK signaling.

To determine whether upd3 and JNK signaling are required for oenocyte PIS-induced cardiac dysfunction, we analyzed the genetic interaction between *Pex5* and *upd3* (or *kay*, *Jra*) in nonautonomous regulation of cardiac arrhythmia. KD of either *upd3*, or *kay*, or *Jra* specifically in oenocytes attenuated *Pex5* KD-induced cardiac arrhythmia (Fig. 5j, k). *upd3* KD alone did not affect cardiac arrhythmicity (Fig. 5l). Thus, these data confirm that upd3 is the primary factor produced in response to oenocyte-specific PIS and JNK activation to mediate oenocyte–heart communication.

Patients with Zellweger syndrome, due to the mutations in PEXs (e.g., PEX1 and PEX5), often develop severe hepatic dysfunction and show elevated inflammation[44]. To test whether peroxisomal dysfunction in patients with Zellweger syndrome also induces the production of proinflammatory cytokines, we measured the expression of *IL-6* and the phosphorylation of JNK in PEX1-G843D-PTS1 cell line. This cell line is a human fibroblast cell line that was isolated, transformed, and immortalized from patients with PEX1-p.G843D allele. Similar to *Pex5* KD flies, PEX1-G843D-PTS1 cells showed significant induction of *IL-6* transcripts (Fig. 5m) and elevated phospho-JNK levels (Fig. 5n, o). We further examined the effects of *PEX1$^{G843D}$* mutations on the production of $H_2O_2$ and peroxidase activity, instead of general ROS, using Amplex Red Assay Kit. Consistently, *PEX1$^{G843D}$* mutant cells showed elevated $H_2O_2$ levels (Supplementary Fig. 5k). Together, our data suggest an evolutionarily conserved mechanism for PIS-mediated induction of proinflammatory cytokines and activation of JNK signaling.

**Peroxisomal import function is disrupted in aged oenocytes.** Previous studies by us and others show that genes involved in peroxisomal import are downregulated during aging[12,15], suggesting an age-dependent impairment of peroxisomal import function. To monitor peroxisomal import during oenocyte aging, we first performed immunostaining using an anti-SKL antibody that recognizes peroxisomal matrix proteins containing the peroxisome targeting sequence (PTS), an SKL tripeptide sequence. We found that the number of SKL-positive punctae were significantly reduced in aged oenocytes, suggesting that the import of endogenous peroxisomal matrix proteins was impaired at old ages (Fig. 6a, b). However, the reduced anti-SKL immunostaining might be caused by the decreased expression of the peroxisomal matrix proteins during aging[12,15]. To address this confounding effect, we directly examined peroxisomal import using a peroxisomal reporter in which YFP is engineered with a C-terminal PTS sequence (YFP-PTS). We transiently induced YFP-PTS

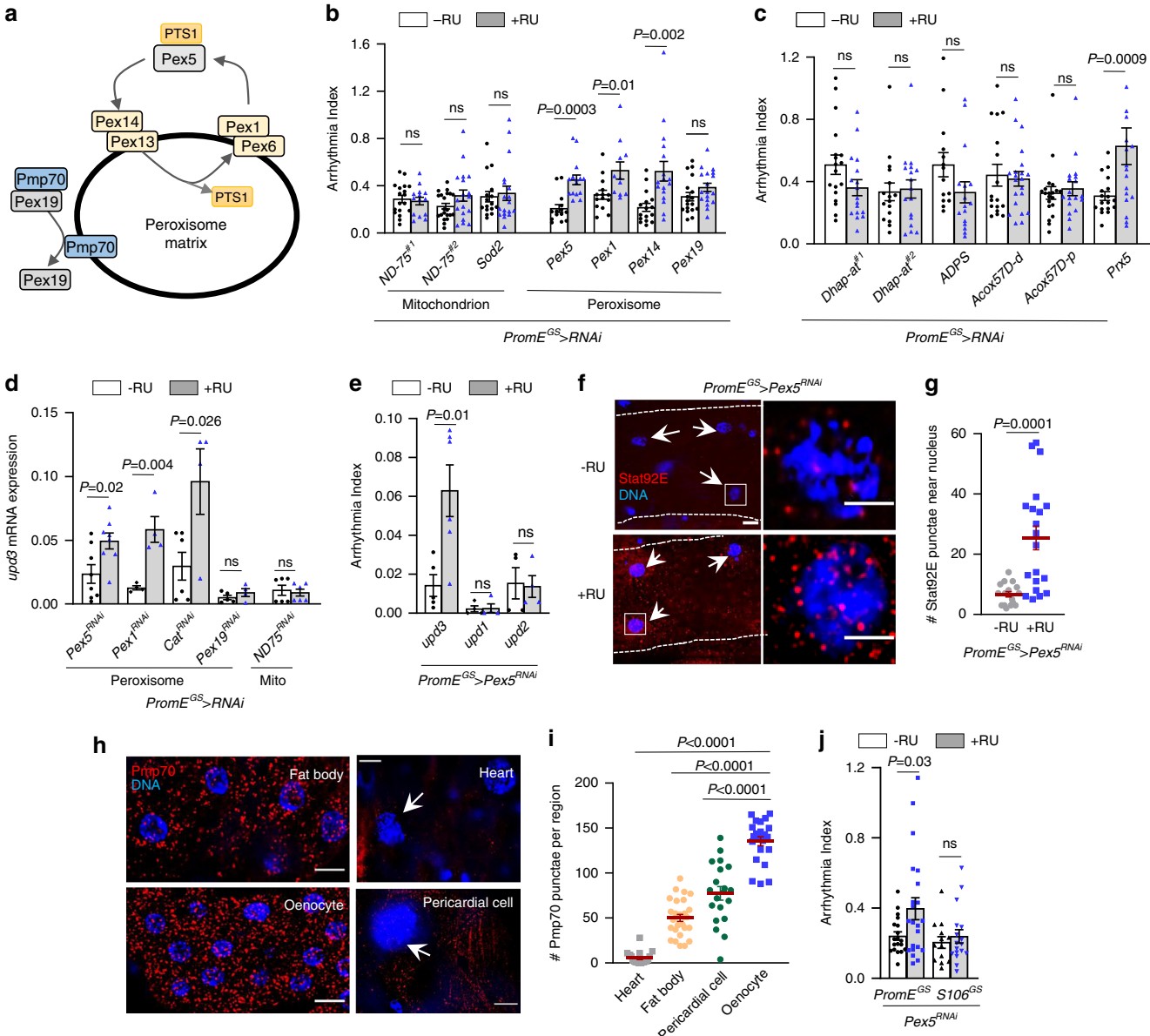

**Fig. 4 Impaired peroxisomal import in oenocytes induces *upd3* and promotes cardiac arrhythmia. a** Schematic diagram showing the key genes involved in peroxisomal import and membrane assembly. **b** Arrhythmia index of oenocyte-specific knockdown of mitochondrial complex I subunit *ND-75* (two independent RNAi lines), mitochondrial Mn superoxide dismutase *Sod2*, peroxisomal import factors (*Pex5, Pex1, Pex14*), and peroxisomal membrane assembly factor (*Pex19*), ($n_{left-right}$ = 14, 15, 14, 11, 16, 17, 18, 18 flies). Oenocyte-specific GeneSwitch driver (*PromE$^{GS}$-Gal4*) was used. **c** Arrhythmia index of oenocyte-specific knockdown of peroxisomal matrix enzymes, *Dhap-at, ADPS, Acox57D-d, Acox57D-p, Prx5* ($n_{left-right}$ = 17, 18, 18, 18, 18, 17, 21, 20, 18, 16, 17, 14, 17 flies). **d** QRT-PCR analysis showing relative mRNA levels of *upd3* from dissected oenocytes of *Pex5, Pex1, Cat, Pex19* and *ND-75* KD flies. Mito: Mitochondrion. N = 4 biological samples from 2 independent experiments. **e** QRT-PCR analysis showing relative mRNA levels of *upd1, upd2, upd3* from dissected oenocytes of oenocyte-specific *Pex5 KD* flies. N = 3 biological samples, from 2 independent experiments. **f** Representative images of Stat92E immunostaining of heart tissues dissected from flies with (+RU) or without (−RU) oenocyte-specific *Pex5* KD. Arrows indicate cardiomyocyte nuclei. White boxes indicate the regions shown in the right insets. Scale bar: 20 μm (inset: 5 μm). **g** Quantification on the number of Stat92E punctae around cardiomyocyte nuclei of oenocyte-specific *Pex5* KD flies (blue dots). Dot plot shows the quantifications of 6 biological replicates, 3-4 selected regions of interest (ROIs) per replicate. **h** Representative images of Pmp70 immunostaining in fly tissues. Arrows indicate cardiomyocyte and pericardial cell nuclei. Scale bar: 6.7 μm. **i** Quantification of Pmp70-positive peroxisomes per region of interest. Dot plot shows the quantifications of 6 biological replicates, 4 ROIs per replicate. **j** Arrhythmia of flies with *Pex5* KD in either oenocytes (*PromE$^{GS}$-Gal4*) or fat body/gut (*S106GS-Gal4*) ($n_{left-right}$ = 19, 22, 13, 17 flies). Data are represented as mean ± SEM. *P* values are calculated using two-sided unpaired t-test (**g**), one-way ANOVA followed (**i**), or two-way ANOVA (**b**–**e**, **j**) by Holm-Sidak multiple comparisons. ns: not significant.

reporter expression in young and aged oenocytes using *PromE$^{GS}$-Gal4* (with 1-day RU486 feeding). Consistent with the results from anti-SKL immunostaining, aged oenocytes showed less YFP-PTS punctae, which indicates that fewer YFP-PTS proteins were imported into peroxisomes in aged flies (Fig. 6c).

To further confirm whether the YFP-PTS punctae are indeed localized with peroxisomes, we performed colocalization analysis between YFP-PTS and peroxisome marker Pmp70. We found that the number of YFP-PTS punctae after normalized to Pmp70 signals was significantly reduced in aged oenocytes

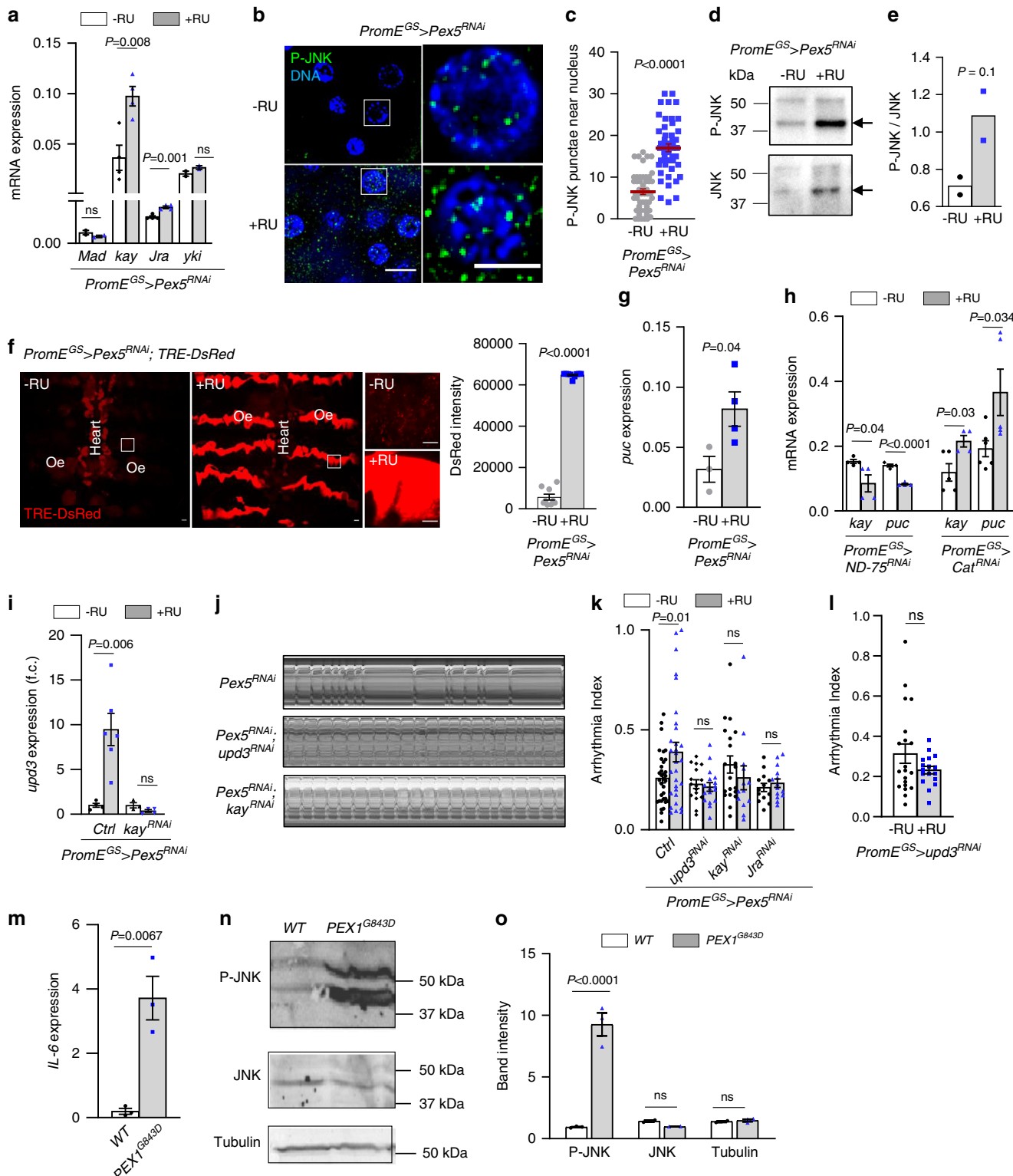

(Fig. 6e), while the number of Pmp70-positive peroxisomes did not change during oenocyte aging (Fig. 6f). Consistently, the colocalization of Pmp70 and YFP-PTS was significantly reduced in aged oenocytes according to the line scan analysis (Fig. 6d) and Pearson's correlation measurement (Fig. 6g). Although the number of peroxisomes (Pmp70-positive) remained unchanged during aging, there were higher number of peroxisomes that showed either none or reduced YFP-PTS signals (Fig. 6c, d). The

peroxisomes with no PTS-positive matrix proteins are known as peroxisomal ghosts (or nonfunctional peroxisomes), which is one of the cellular hallmarks of Zellweger Syndrome[45,46]. Together, these results suggest that peroxisomal import is impaired in aged oenocytes.

Similarly, the decreased peroxisomal import was observed in oenocyte-specific *Pex5* KD flies (Fig. 6h). The KD of *Pex5* significantly reduced the number of SKL-positive punctae and

**Fig. 5 Pex5 KD-mediated PIS induces *upd3* through JNK signaling. a** Expression of *Mad, kay, Jra, yki* in oenocytes of *PromE^GS>Pex5^RNAi* flies, n = 2 biological samples from 2 independent experiments. **b** P-JNK immunostaining of *Pex5* KD oenocytes with (+RU, blue dots) or without (−RU, grey dots). Scale bar: 6.7 µm. **c** Quantification of **b** N = 6 biological replicates (8 nuclei per replicate). **d** Western blots showing the levels of P-JNK and JNK (arrows) in control (−RU) and *Pex5* KD (+RU) oenocytes. **e** Quantification of **d** N = 2 independent biological samples. **f** Fluorescence imaging of JNK reporter TRE-DsRed from control (−RU) and oenocyte *Pex5* KD (+RU). Scale bar = 20 µm. Quantification is shown on the right. N = 9 flies. **g** Expression of *puc* in oenocytes from flies with *Pex5* kd oenocyte. N = 4 independent biological samples. **h** Expression of *kay* and *puc* in oenocytes dissected from flies with oenocyte-specific knockdown of *ND-75* and *Cat*. N = 4 independent biological samples. **i** Expression of *upd3* in oenocytes dissected from flies with *Pex5* KD or *Pex5; kay* double KD. N = 4 biological samples. Ctrl genotype is *UAS-Pex5^RNAi/attP40; PromE^GS-Gal4/+*. + indicates wild type. **j** Representative M-mode of flies with oenocyte *Pex5* KD, *Pex5; upd3* double KD, or *Pex5; kay* double KD. **k** Arrhythmia of flies with oenocyte *Pex5* KD, or *Pex5; upd3* double KD, or *Pex5; kay* double KD, or *Pex5; Jra* double KD. Ctrl genotype is the same as **i** ($n_{left-right}$ = 34, 30, 16, 17, 20, 16, 20, 13, 13, 14 flies). **l** Arrhythmia of flies with oenocyte upd3 KD ($n_{left-right}$ = 20, 17). **m** Expression of *IL-6* in human PEX1-G843D-PTS1 cells. N = 3 independent biological samples. **n** Western blots showing P-JNK and JNK in wild-type and human PEX1-G843D-PTS1 cells. **o** Quantification of Panel **n**. nJNK/tubulin = 2, nPJNK = 3 biological samples. Data are represented as mean ± SEM. *P* values are calculated using two-sided unpaired t-test (**c**, **e**–**g**, **l**, **m**), two-way ANOVA followed by Holm-sidak multiple comparisons (**a**, **h**, **i**, **k**, **o**) ns: not significant.

colocalization between SKL and Pmp70 (Fig. 6j, l), while the number of Pmp70-positive peroxisomes was only weakly reduced (Fig. 6k). In contrast, the number of SKL containing punctae in *Pex19* KD oenocytes remained unchanged (Fig. 6i, j), while *Pex19* KD significantly decreased the number of Pmp70-positive peroxisomes (Fig. 6k). These results suggest that Pex5, but not Pex19, plays an important role in peroxisomal import, which may explain why *Pex5* KD, but not *Pex19* KD, induces ROS, *upd3* production, and cardiac dysfunction,

**Oenocyte *Pex5* activation alleviates PIS and arrhythmia in aging**. Although peroxisomal import function declines with age, the causal relationship between impaired peroxisomal import and tissue aging remains elusive. To determine the physiological significance of peroxisomal import in aging and cardiac health, we next asked whether preserving peroxisomal import function in oenocytes could attenuate the production of proinflammatory cytokines and protect hearts from PQ- and aging-induced arrhythmicity. Our previous oenocyte translatomic profiling showed that *Pex5* is downregulated upon PQ treatment and aging[12]. We thus used a CRISPR/Cas9 transcriptional activation system[47] to drive physiologically relevant expression of *Pex5* from its endogenous locus. To express *Pex5* specifically in oenocytes, we first combined a catalytically dead Cas9 line (*UAS-dCas9-VPR*) with oenocyte-specific driver line (*PromE^GS-Gal4*), and then cross it with flies expressing *Pex5* guide RNA. We predict that overexpression of *Pex5* in oenocytes could rescue impaired peroxisomal import, block age-related induction of *upd3*, and preserve cardiac health.

As expected, *Pex5* transcripts in oenocytes were induced about twofold using the *dCas9-VPR* system (Fig. 7a). Interestingly, *Pex5* overexpression attenuated the age-related *upd3* induction in aged oenocytes (Fig. 7b). The overexpression of *Pex5* also restored peroxisomal import function in aged oenocytes, as indicated by higher number of anti-SKL immunostaining (functional peroxisomes) (Fig. 7c, d) and lower percentage of peroxisomal ghosts (nonfunctional peroxisomes) (Fig. 7c, e). *Pex5* overexpression did not change the total peroxisome numbers (Pmp70-positive punctae) (Fig. 7c, f). Our result is consistent with the previous study showing that overexpression of *Pex5p* partially restores the peroxisomal import function of *pex14Δ* mutants in yeast[48]. Intriguingly, oenocyte-specific overexpression of *Pex5* blocked PQ-induced arrhythmicity (Fig. 7g, h), and preserved cardiac function at old ages (Fig. 7i, j). Besides protecting cardiac function, oenocyte-specific overexpression of *Pex5* alleviated age-related ROS production in oenocytes (Fig. 7k, l). Together, our data showed that preserving peroxisomal import function via overexpression of endogenous *Pex5* can block the production of ROS and peroxikine *upd3*, and protect hearts from oxidative

stress- and aging-induced cardiomyopathy (Fig. 7m). Thus, our studies provide strong evidence suggesting that age-related impairment of peroxisomal import is an understudied cause of tissue aging.

## Discussion

Here our studies provide direct evidence linking oenocyte/liver dysfunction to cardiac health. We also demonstrate that impaired peroxisomal import at old ages is the cause of inflammaging, whereas maintaining peroxisomal import slows tissue aging. Our genetic analyses show that aging-induced proinflammatory cytokine upd3 from oenocytes/liver, in response to impaired peroxisomal import, activates JAK-STAT signaling in the heart to induce cardiac arrhythmia. Interestingly, either reducing *upd3* or overexpressing *Pex5* in oenocytes alleviates aging- or PQ-induced cardiac dysfunction. Together, our studies established the vital role of hepatic peroxisomal import in activating systemic inflammation and nonautonomous regulation of cardiac aging. Our findings suggest that protecting oenocyte peroxisome function is critical for prolonging cardiac healthspan.

IL-6 is the most important proinflammatory cytokine that is associated with inflammaging and age-related diseases[4]. The upregulation of IL-6 is often seen upon tissue injury, such as ischemia-reperfusion[49]. IL-6 is also found to be secreted as a myokine from skeletal muscle during exercise[50] to mediate the beneficial effects. However, the studies using IL-6 knockout mouse models suggest that IL-6 is pathogenic and proinflammatory. For example, IL-6-deficient mice show reduced expansion of CD4+ T cells and autoimmune response[51]. In *Drosophila*, IL-6-like cytokine upd3 plays a key role in intestinal stem cell homeostasis[52–54], reproductive aging[55], glucose homeostasis, and lifespan[24]. Although unpaired family members, especially upd2, have been shown to relays nutrient signals from FB to brain[56], the role of upd3 in intertissue communication and systemic aging regulation has not been fully established. It is not known which tissue is the main source of upd3 production, and which tissues upd3 targets during normal aging. In the present study, we show that upd3, but not upd1 and upd2, mediates the communication between oenocytes and hearts. Among all tissues tested, oenocytes produce highest amount of upd3 at old ages. Interestingly, oenocyte-specific upd3, not gut/adipocyte, modulates cardiac function during aging. Thus, our studies demonstrate that oenocytes are the main source of upd3 in aged animals. It remains to be determined why upd3 produced from different tissues exhibits distinct roles, and whether oenocyte-produced upd3 can target tissues beyond the heart (e.g., the central nervous system).

It is well known that mitochondria play an important role in regulating inflammaging[4]. Mitochondrial stresses (such as mitochondrial unfolded protein response, UPR^mt) can lead to

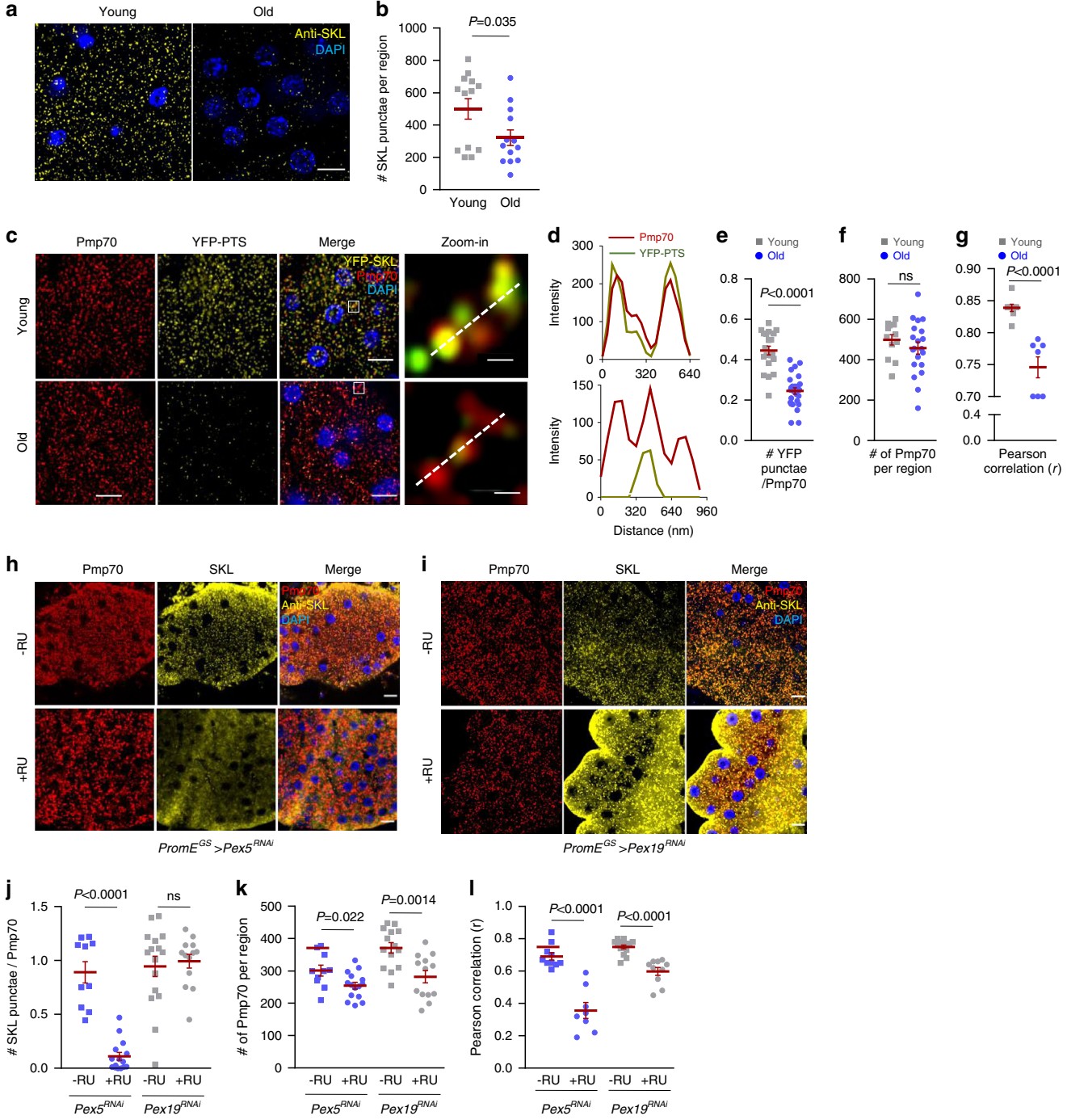

**Fig. 6 Peroxisomal import function is disrupted in aged oenocytes. a** Representative images of anti-SKL immunostaining of young and aged wild-type oenocytes. Scale bar: 6.7 μm. **b** Quantification on the number of SKL-positive punctae in **a**. Dot plot shows the quantifications of 6 biological replicates, 2 ROIs per replicate. **c** Representative images to show co-localization of Pmp70 and YFP-PTS in young and aged oenocytes (Scale bar: 6.7 μm). Insets on the right show zoom-in peroxisome structures (the regions indicated by the white boxes in the merged panels, Scale bar: 125 nm). **d** Line scan analysis to show the fluorescence intensity of Pmp70 (red) and YFP-PTS (green) crossing peroxisomes in the insets (dashed line). **e** Quantification of the number of YFP-positive punctae normalized Pmp70 in **c**. N = 6. Dot plot shows the quantifications of 6 biological replicates, 4 ROIs per replicate. **f** Quantification of the number of Pmp70-positive punctae in **c**, N = 6 biological replicates, 4 ROIs per replicate. **g** Pearson correlation quantification measuring the correlation coefficient of the co-localization between Pmp70 and YFP-PTS in **c**. N = 6 biological replicates. **h** Representative images of anti-SKL and anti-Pmp70 immunostaining of *PromE^GS^>Pex5^RNAi^* oenocytes. Scale bar: 6.7 μm. **i** Representative images of anti-SKL and anti-Pmp70 immunostaining of *PromE^GS^>Pex19^RNAi^* oenocytes. Scale bar: 6.7 μm. **j** Quantification of the number of SKL-positive punctae normalized Pmp70 in **h**, **i**. **k** Quantification of the number of Pmp70-positive punctae in **h**, **i**. **l** Quantification of Pearson correlation coefficient of the co-localization between Pmp70 and SKL in **h**, **i**. 6 biological replicates, 3 ROIs per replicate. Data are represented as mean ± SEM. *P* values are calculated using two-sided unpaired t-test (**b**, **e–g**). or two-way ANOVA by Holm-Sidak multiple comparisons (**j–l**), ns: not significant. For specific statistical number, please refer to the source data.

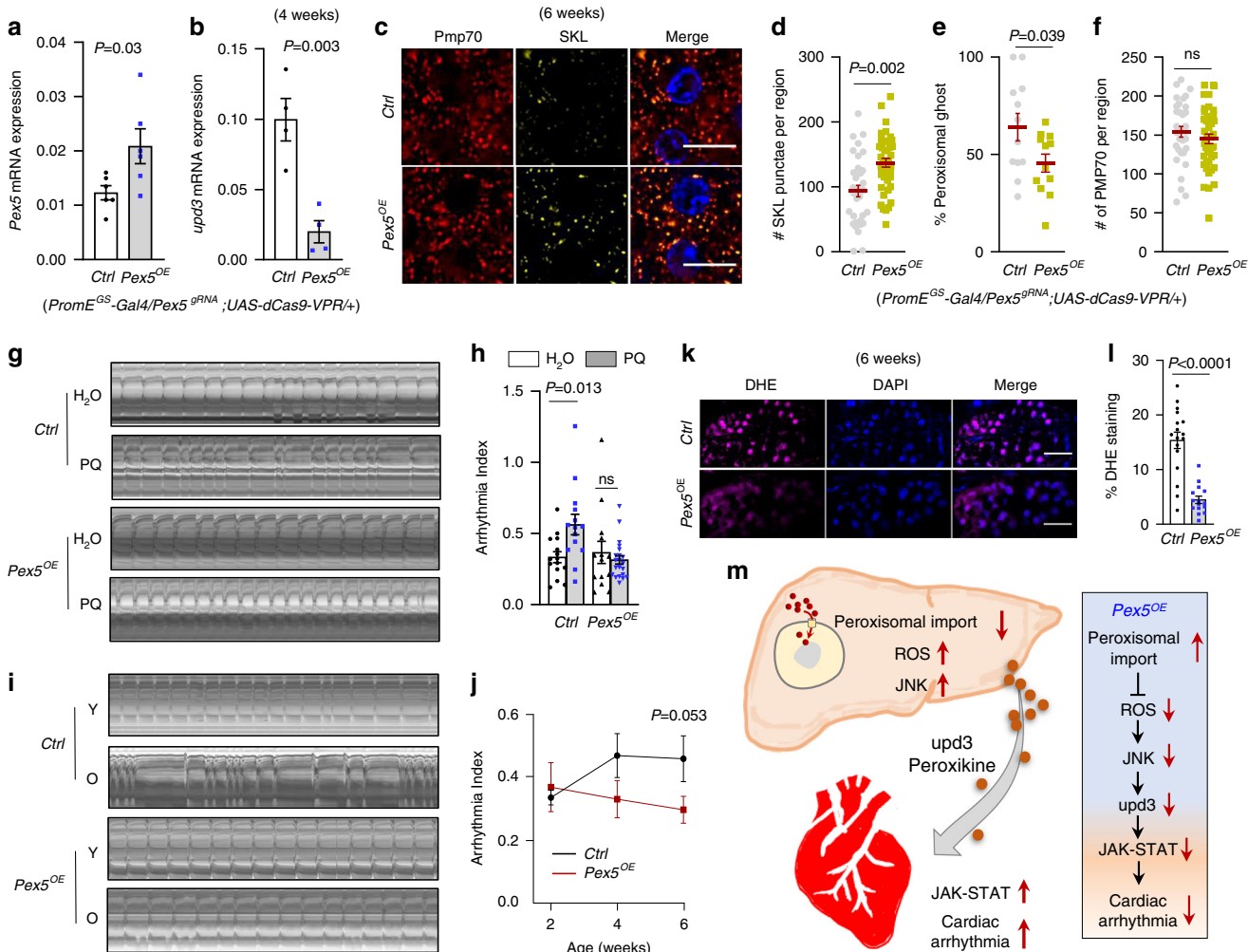

**Fig. 7 Oenocyte-specific *Pex5* activation alleviates age-related PIS and preserves cardiac function. a** QRT-PCR analysis showing expression of *Pex5* in oenocytes dissected from flies with oenocyte-specific overexpression of *Pex5*. N = 3 biological samples from 2 independent experiments. **b** QRT-PCR analysis showing relative mRNA expression of *upd3* in 4-week-old oenocytes dissected from flies with oenocyte-specific overexpression of *Pex5*. N = 4 independent biological samples. **c** Representative images to show co-localization of anti-Pmp70 and anti-SKL immunostaining in 6-week-old oenocytes dissected from flies with oenocyte-specific overexpression of Pex5 (yellow dots). Scale bar: 6.7 μm. **d** Quantification of SKL-positive punctae in **c**. Quantification of the percentage of peroxisomal ghosts (Pmp70-punctae with no SKL signals) per region of interest (166.41 μm2) in **c**. **f** Quantification of Pmp70-positive punctae in **c**. **d**–**f**. Dot plot shows the quantifications of 6 biological replicates, 6 ROIs per replicate. **g** Representative M-mode of flies with oenocyte-specific *Pex5* overexpression under paraquat treatment. **h** Arrhythmia index of flies with oenocyte-specific *Pex5* overexpression under paraquat treatment. ($n_{left-right}$ = 15,14,14,21 flies). **i** Representative M-mode of flies with oenocyte-specific *Pex5* overexpression during normal aging. **j** Arrhythmia index of flies with oenocyte-specific *Pex5* overexpression (red dots) during normal aging (2-week, 4-week and 6-week-old), $n_{ORU,y-o}$ = 35, 15, 15 flies; $n20_{ORU,y-o}$ = 9, 15, 18 flies. **k** DHE staining in 6-week-old oenocytes from flies with oenocyte-specific overexpression of *Pex5*. Scale bar: 20 μm. **l** Quantification of the percentage of DHE-positive staining in **k**. N = 6 flies, 3 ROIs per replicate. Ctrl genotype is *PromE^{GS}>Pex5^{OE}* with no RU feeding. Data are represented as mean ± SEM. *P* values are calculated using two-sided unpaired t-test (**a**, **b**, **d**–**f**, **l**, **j**) or two-way ANOVA, followed by Holm-sidak multiple comparisons (**h**). **m** Proposed model to show that impaired hepatic peroxisomal import promotes ROS production, JNK activation, and peroxikine upd3 expression, which non-autonomously controls cardiac JAK-STAT and arrhythmia. Overexpression of endogenous *Pex5* preserves cardiac health.

the induction and secretion of hormonal factors, termed mito-kines, which in turn activate mitochondrial stress responses in distant tissues[57,58]. There is a strong interplay between mito-chondria and peroxisomes in the maintenance of intracellular redox homeostasis[59]. In fact, peroxisome deficiency can alter mitochondrial morphology and respiration[60,61], and induce mitochondria-mediated cell death[62]. However, the role of per-oxisome in inflammaging still remains elusive. In the present study, we discover that KD of peroxisomal import receptor *Pex5* triggers a significant production of cytokine upd3 in a JNK-dependent manner. Although it remains to be solved how impaired peroxisomal import activates JNK signaling

pathway. Interestingly, PIS in both flies and human fibroblasts of Zellweger spectrum patients activates JNK pathway and the production of IL-6-type cytokines. These findings suggest the existence of a conserved cellular stress response to impaired peroxisomal import. Similar to the production of mitokines upon mitochondrial stresses, impaired peroxisomal import can also induce the expression and release of a group of inflam-matory factors (such as upd3/IL-6) to modulate cellular function of distant tissues. We term PIS-induced hormonal factors per-oxikines. The peroxikines play essential roles in maintaining tissue homeostasis upon PIS, since we can modulate the pro-duction of these factors in one tissue to protect distant tissues.

The expression of the peroxikines could potentially serve as biomarkers for cellular PIS and inflammaging.

Age-related impairment of peroxisomal import has been previously reported in human senescence cells and nematodes[15–17], although the underlying mechanism is poorly understood. Possible mechanisms include increased Pex5 oxidation and degradation[63], dampened Pex5 recycling activity[17], decreased ATP production[64], and decreased Pex5 expression during aging[12]. Our data suggest that functional Pex5 are significantly reduced during normal aging, which explains why overexpressing *Pex5* can restore the import function and delay age-related pathologies. It should be noticed that excess expression of *PEX5* can block the normal import function[65]. We utilized a CRISPR/Cas9 activation system to achieve optimal expression of endogenous *Pex5*, which has proven to be an effective way to restore impaired peroxisomal import function and attenuate aging-induced cardiac dysfunction. Thus, our genetic analysis of Pex5 establish peroxisomal import as a cause of tissue aging.

Besides the regulation of redox homeostasis, peroxisome performs several other essential functions, such as fatty acid beta oxidation and ether phospholipid biosynthesis. Although our studies cannot fully exclude the role of peroxisomal beta oxidation and ether phospholipid biosynthesis in oenocyte–heart communication and cardiac aging, impaired peroxisomal import could also lead to the accumulation of very long-chain fatty acids and reduced production of anti-inflammatory lipids, such as docosahexaenoic acids (DHA)[66]. Thus, the decreased of DHA production, due to impaired peroxisomal import, can promote chronic inflammation in the liver and induce cardiomyopathy at old ages. Further studies are needed to carefully examine these possibilities.

In summary, our studies reveal that *Drosophila* oenocytes (hepatocyte-like tissue) play a vital role in nonautonomous regulation of cardiac aging. In response to PIS, oenocytes produce proinflammatory peroxikine upd3, to modulate heart function. Protecting oenocytes from PIS and upd3 production can alleviate cardiomyopathy under oxidative stress and aging. Our findings suggest that peroxisome is a central regulator of inflammaging and intertissue communication. Future work will be of interests to examine the role of liver peroxisomes in age-related cardiac diseases in mammalian systems.

## Methods

Detailed reagent information is provided in Supplementary Table 1.

**Drosophila husbandry and strains**. A detailed list of fly strains is provided in Supplementary Table 1. The following genotypes were used as control in the KD or overexpression experiments: *yw^R*, *w^1118*, *Gal4^RNAi* (BDSC#35783), *y^1 v^1*; *P[CaryP] attP2* (BDSC #36303), or *y^1 v^1*; *P[CaryP]attP40* (BDSC # 36304). Two independent RNAi lines were used in the KD of *upd3*, *Pex5*, and *ND-75*.

Female flies were used in all experiments. Flies were maintained at 25 °C, 60% relative humidity, and 12-h light/dark cycle. Adults and larvae were reared on a standard cornmeal and yeast-based diet, unless otherwise noted. The standard cornmeal diet consists of the following materials: 0.8% cornmeal, 10% sugar, and 2.5% yeast. RU486 (mifepristone, Fisher Scientific) was dissolved in 95% ethanol, and added to standard food at a final concentration of 100 μM for all the experiments except for *Pex5* overexpression, where 200 μM of RU486 was used. For the use of GeneSwitch driver, gene KD or overexpression was achieved by feeding flies on RU486 food for 5 days, unless otherwise noted. For PQ feeding assay, PQ (dichloride hydrate pestanal, Sigma) was dissolved in distilled water to a final concentration of 10 mM (Figs. 1–3) or 20 mM (Figs. 4–7). About 150 μl of PQ working solution was prepared freshly and added onto the surface of fly food. PQ solution was air dried for 20 min at room temperature prior to use. In Fig. 3c, flies were fed on either 5% sucrose or 5% sucrose plus 20 mM of PQ on the agar vials. Fly heart tissues were dissected after 24 h of feeding.

**Human PEX1-G843D-PTS1 cell culture**. The human PEX1-G843D-PTS1 cell lines were from the Braverman Laboratory[67]. It is a fibroblast cell line that was originally isolated, transformed, and immortalized from patients with PEX1-p. G843D, a missense allele that accounts for one-third of all Zellweger spectrum

disorder alleles. It stably expresses GFP-PTS1 reporter. The PEX1-G843D-PTS1 cell lines were cultured in DMEM with 10% FBS at 5% $CO_2$. The wild-type cell lines were established from the fibroblasts of healthy donors. These experiments were conducted under ethical approval obtained for research through the REB at Kennedy Krieger Institute, Baltimore, MD, to the Peroxisome Disease Laboratory, under the grant General Clinical Research Centres (RR0052 and RR00722) (PEX1-G843D-PTS1 cell lines), and an IRB protocol to the Eric Shoubridge Laboratory at the Montreal Children's Hospital (wild-type cell line). Written informed consent was obtained from involved subjects.

**Fly heartbeat analysis**. Cardiac contractility was measured using semi-intact female *Drosophila*[68]. Female flies were dissected in oxygenated artificial hemolymph to measure heart contractions within the abdomen. Artificial adult hemolymph contains 108 mM $NaCl_2$, 5 mM KCl, 2 mM $CaCl_2$, 8 mM $MgCl_2$, 1 mM $NaH_2PO_4$, 4 mM $NaHCO_3$, 15 mM 4-(2-hydroxyethyl)-1-piperazineethanesulfonic acid (HEPES), 10 mM sucrose, and 5 mM trehalose, at pH 7.1. High speed movies of heart contraction (30 s, 100 frames/s) were recorded using a Hamamatsu ORCA-Flash 4.0 digital CMOS camera (Hamamatsu) on an Olympus BX51WI microscope with a 10× water immersion lens (Olympus). The live images taken from the heart tub within abdominal A3 segment were processed using HCI imaging software. M-modes and cardiac parameters were generated using SOHA, a MATLAB-based image application. A 15 s of representative M-mode was presented in the figures to show the snapshot of the movement of heart wall over time. AI is calculated as the standard deviation of all heart periods in each fly normalized to the median heart period. Heart period is the pause time between the two consecutive diastole[68].

**DHE staining**. ROS detection was performed using DHE dye (Fisher Scientific). Female oenocytes were dissected in 1x PBS according to a published protocol[69] with modification. Dorsal side of abdomen of female flies are placed on Vaseline, spread evenly on a culture dish. After removing unwanted organs, such as ovaries and intestine, abdominal FB was removed by liposuction and fly carcass with intact oenocytes was incubated in 30 μM of DHE solution for 5 min in the dark. After washing with PBS three times, the oenocytes were stained with Hoechst 33342 (1 μg/ml) (ImmunoChemistry Technologies) for 10 min, mounted in ProLong Gold Antifade Reagent (Thermo Fisher Scientific), and imaged with an epifluorescence-equipped BX51WI microscope (Olympus).

**Measurement of $H_2O_2$ by amplex red**. The human PEX1-G843D-PTS1 cells were homogenized in PBS and clarified by centrifugation. $H_2O_2$ amounts in the resultant supernatants were measured using the Amplex Red Hydrogen Peroxide/Peroxidase Assay Kit (Thermo Fisher) and normalized to protein amounts. Protein amounts were measured using a Qubit II Fluorometer (Thermo Fisher). Experiments were done in triplicate.

**Secretory factor screen**. *Drosophila* secretory proteins were first identified from the Gene List Annotation for *Drosophila*[23] and compared with our recent oenocyte translatomic analysis[12]. Candidate genes that are differentially expressed in aged or PQ-treated oenocytes were selected in a RNAi screening using oenocyte-specific driver *PromE-Gal4*. About 20 mated females (3–5-day old) per genotype were treated with 10 mM PQ (as described above) for 24 h before heartbeat analysis (SOHA).

**RNA extraction and QRT-PCR**. Adult tissues (oenocyte, heart, gut, FB, pericardial cells) were all dissected in 1 × PBS before RNA extraction. For oenocyte dissection, we first removed FB through liposuction and then detached oenocytes from the cuticle using a small glass needle. Tissue lysis, RNA extraction, and cDNA synthesis were performed using Cells-to-CT Kit (Thermo Scientific).

QRT-PCR was performed with a Quantstudio 3 Real-Time PCR System and PowerUp SYBR Green Master Mix (Thermo Fisher Scientific). Two to three independent biological replicates were performed with two technical replicates. The mRNA abundance of each candidate gene was normalized to the expression of *RpL32* for fly samples and *GAPDH* for human samples, by the comparative $C_T$ methods. Primer sequences are listed in the Supplementary Table 2.

**Antibody and immunostaining**. The following antibodies were used in immunostaining: anti-GFP (Cell Signaling Technology, #2956S, 1:200), anti-P-JNK (Thr183/Tyr185) (Cell Signaling Technology, #4668S, 1: 100), anti-Stat92E (1:900, a gift from Steven X. Hou), anti-Pmp70 rabbit polyclonal antibody (1:200, generated against the *Drosophila* Pmp70 C-terminal region 646-665 (DGRGSYEFA-TIDQDKDHFGS) by Pacific Immunology), and anti-SKL rabbit polyclonal antibody (1:250, raised by Richard Rachubinski). An anti-Pmp70 Guinea Pig polyclonal antibody (a gift from Kyu-Sun Lee, 1:500) was used in peroxisome import assay (Fig. 7). Secondary antibodies were obtained from Jackson ImmunoResearch.

Adult tissues were dissected and fixed in 4% paraformaldehyde for 15 min at room temperature. Tissues were washed with 1x PBS with 0.3% Triton X-100 (PBST) for three times (~5 min each time), and blocked in PBST with 5% normal goat serum for 30 min. Tissues were then incubated overnight at 4 °C with primary antibodies

diluted in PBST, followed by the incubation with secondary antibodies for 1 h at room temperature. After washes, tissues were mounted using ProLong Gold antifade reagent (Thermo Fisher Scientific) and imaged with an epifluorescence-equipped BX51WI microscope and an FV3000 Confocal Laser Scanning Microscope (Olympus). DAPI or Hoechst 33342 was used for nuclear staining.

**Image analysis and quantification**. Fluorescence images were first processed and deconvoluted using Olympus CellSens Dimension software (Olympus). The number of punctae or fluorescent intensity/area in a selected region of interest (ROI, 918.33 $\mu m^2$ for Figs. 5 and 6, 367.29 $\mu m^2$ for Figs. 4 and 7) was measured using the CellSens Measure and Count module after adjusting the intensity threshold to remove background signals. To quantify the punctae near the nucleus, we first selected ROIs surround the nucleus according to Hoechst signal, and then counted the punctae number within each ROI using the CellSens Measure and Count module. Two to four ROIs were analyzed for each image. The imaging quantifications were done single or double blind.

Colocalization analysis was performed in ImageJ. Coloc 2 plugin function was used to calculate Pearson's correlation ($r$). Line scan analysis in Fig. 6 was performed by creating composite images containing SKL and Pmp70 immunostaining. We then drew a line crossing three peroxisomes in a selected ROI, and intensity plots were generated using multi-channel plot profile in the BAR package (https://imagej.net/BAR).

To identify peroxisomal ghost, images were processed, and thresholds were adjusted following the procedures described above. Image segmentation was performed for each channel (SKL: green, Pmp70: red) using Otsu thresholding. Then SKL and Pmp70 channels were merged in ImageJ. For each merged image, three ROIs (166.41 $\mu m^2$) were selected and the total number of peroxisome and peroxisomal ghosts (Pmp70-positive punctae with no SKL signals) were manually counted. The percentage of peroxisomal ghosts was presented.

**Hemolymph extraction**. Hemolymph was extracted by piercing into the thoraces of 30 adult female flies (each genotype) with glass needles made of capillary tubes using Sutter Puller (Sutter Instrument, Model P-97). The following settings were used to prepare glass needle: heat = 302, pull = 75, vel = 75, time = 155, $P$ = 435. Vacuum was used to facility the hemolymph extraction. About 0.5 $\mu$l of hemolymph extracted from 30 flies was pooled in a 1.5 ml microcentrifuge tube containing 10 $\mu$l of 1x PBS and protein inhibitor cocktail. After centrifuge at $3000 \times g$ for 2 min at 4 °C, the hemolymph samples were snap-frozen with liquid nitrogen and stored at −80 °C for western blot analysis. Equal amount of hemolymph (0.5 $\mu$l in 10 $\mu$l of 1x PBS) was denatured and loaded on SDS-PAGE gels.

**Western blot analysis**. Proteins samples were denatured with Laemmli sample buffer (Bio-Rad, Cat# 161-0737) at 95 °C for 5 min. Then proteins were separated by Mini-PROTEAN® TGX Precast Gels (Bio-Rad). Following incubation with primary and secondary antibodies, the blots were visualized with Pierce ECL Western Blotting Substrate (Thermo Scientific). The following antibodies were used: anti-GFP (Cell Signaling Technology, #2956S, rabbit, 1:1000), anti-P-JNK (Cell Signaling Technology, #9255, 1:2000), anti-JNK (Cell Signaling Technology, #9252, 1:1000), and anti-Tubulin (Sigma, #T5168, 1:2000). For hemolymph samples, 4–15% Mini-PROTEAN® TGX Stain-Free™ Precast Gels (Bio-Rad, Cat# 456-8085) was used. The Stain-Free™ gel was UV activated (Bio-Rad) using Chemi-Doc MP Imager to visualize total protein loading before immunoblotting.

**Statistical analysis**. GraphPad Prism (GraphPad Software, version 6.07) was used for statistical analysis. Unpaired two-tailed Student's $t$ test or one-way ANOVA (Holm–Sidak comparison) was performed to compare the mean value between control and treatment groups. The effects of mutants during aging or oxidative stress were analyzed by two-way ANOVA, followed by Holm–Sidak multiple comparisons. The outliers were excluded using robust regression and outlier removal method (Q = 1%) prior to the data analysis. Outlier values can be found in the Source Data file.

**Reporting summary**. Further information on research design is available in the Nature Research Reporting Summary linked to this article.

## Data availability

Full scans of the gels and blots are available in Supplementary Fig. 6. Primary data of interest are provided in the Source Data file of this paper (Figs. 1, 2b–k, and 3–7 and Supplementary Figs. 1–5). RNA-seq results on aging and PQ treated oenocytes (Fig. 2a) were previously deposited to NCBI 's Gene Expression Omnibus (GEO) (Accession # GSE112146). To review GEO files: please go to https://www.ncbi.nlm.nih.gov/geo/query/acc.cgi?acc=GSE112146. All remaining data are available from the corresponding author upon reasonable request. Source Data are provided with this paper.

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

## Acknowledgements

We thank Bloomington Drosophila Stock Center and Harvard Medical School for fly stocks. We thank Steven X. Hou, Richard Rachubinski, and Kyu-Sun Lee for antibody reagents, and Marc Tatar, Alex Gould, Doug Harrison, Heinrich Jasper, Rolf Bodmer, and Erika Bach for fly stocks. We thank Nancy Braverman for *PEX1* mutant cell lines. We also thank Rolf Bodmer and Karen Ocorr for help with fly heartbeat analysis. This work was supported by NIH/NIA R00AG048016, R01AG058741, and AFAR Research Grants for Junior Faculty to H.B., Glenn/AFAR Scholarships for Research in the Biology of Aging to K.H., Alberta Innovates-Collaborative Research and Innovation Opportunities to A.J.S., and Dalhousie Medical Research Foundation to F.D.C.

## Author contributions

K.H. and H.B. conceived the study and wrote the manuscript. K.H. performed most of the experiments. T.M. performed upd3 secretion assay, and K.C. performed SOHA analysis on cardiac JAK-STAT activation. J.K. performed western blots to test the specificity of GFP antibody and p-JNK levels in Pex5 RNAi flies. P.K. performed tissue-specific upd3 expression measurement. Q.J. performed part of the secretory factor screen. F.D.C. performed the analyses on Pex1 patient cells. A.J.S. and F.D.C. guided the design of peroxisomal function analysis and generated Pmp70 antibodies. K.H., F.D.C., and H.B. analyzed the data. All authors reviewed and approved the manuscript.

## Competing interests

The authors declare no competing interests.
