## [Peer Review File · Nature Communications]

Reviewers' comments:

Reviewer #1 (Remarks to the Author):

Huang and colleagues present compelling and interesting data exploring the effects of Oenocyte-mediated inflammation in the age-related decline of cardiac function in *Drosophila*. The authors show that increased oxidative stress in aging oenocytes results in the JNK/Kay-mediated expression of Upd3 and subsequent activation of JAK/STAT signaling in cardiomyocytes, and that this non-autonomous signaling mechanism promotes cardiac arrhythmias. The authors further establish that the cause for the elevated ROS in aging oenocytes is the impairment of peroxisomal import, and, accordingly, show that factors that promote peroxisomal import in oenocytes can rescue age-related cardiac arrhythmias.

The work is expansive and logically presented, the data are overall of good quality and properly interpreted, and the manuscript is well - written and should be of interest to a broad audience. The topic addressed, namely how aging causes systemic inflammation and how dysfunction in specific tissues promotes dysfunction in others is of particular interest to the aging community and the presented data make a significant contribution to our understanding of the loss of homeostasis of the systemic milieu. It is likely that these findings can also be translated to vertebrate/human cases of cardiac dysfunction.

I have a few minor comments that the authors should address before the paper is published:

- The authors use STAT nuclear localization as a readout for STAT activity, but the images are not very clear, and it seems difficult to clearly establish nuclear localization of STAT in oenocytes. There are very good STAT reporters (10XSTAT-GFP and 2XSTAT-GFP) for flies, and the authors should use these to validate their results. Alternatively, one could also use qPCR to establish expression of SOCS36E, a target gene for STAT in flies.
- Figs. 3 H and I are not informative and should be taken out of the manuscript. If STAT activation in cardiomyocytes by oenocyte-expressed Upd3 can be clearly established (see above), then this experiment is unnecessary.
- A critical control for many of the presented data is to show that the PromE and PromEGS drivers are indeed excluded from cardiomyocytes. The authors should show convincing evidence (images or western blots) showing that the drivers are only expressed in oenocytes, not the heart.

Reviewer #2 (Remarks to the Author):

Summary: The submitted article, "Impaired peroxisomal import in *Drosophila* hepatocyte-like cells induces cardiac dysfunction through the pro-inflammatory cytokine Upd3" by Huang et al, uses the *Drosophila* genetic model to examine the possibility that liver-like oenocytes impact cardiac function in the fly. They convincingly show that aging induced cardiac arrhythmia and paraquat induced ROS stress can be blocked by oenocyte-specific over expression of SOD. They identify age-dependent increases in UPD3 expression in oenocytes and show that the effects on UPD3 are mediated by signaling through the JNK pathway. The authors show UPD3 from oenocytes is secreted into the hemolymph and is taken up by the heart. They also show that UPD3 induces JAK-STAT signaling in the heart. Consistent with their observations, oenocyte-specific KD of UPD3 prevented both the age-dependent increase in cardiac expression of STAT and cardiac arrhythmias. Their data supports the notion that it is a decrease in peroxisome import via PEX5 and not impaired mitochondrial function in oenocytes that is the cause of the increased UPD3 signaling. These findings are novel and extremely interesting as they suggest that non-cardiac peroxisomes play important non-autonomous roles in heart function via IL6/UPD3-mediated inter-organ communication.

The authors provide a significant amount of good data to support these conclusions. The data are overall compelling but there are some controls that are missing and see comments below.

Major Points-

1. The RU-based Gene induction experiments are good but missing key controls, as feeding with RU by itself has some cardiac effects. The driver lines should be outcrossed to flies from the RNAi or OE genetic backgrounds and treated with and without RU to control for the effect of drug treatment and driver genetic background. Stock centers have the background insertion lines that should be used for these controls, (eg. the appropriate control for Fig. 5f would be a cross between the RNAi background line and the PromE GS line with the progeny being fed +/- RU).

2. For the experiments to identify the mediators of Pex5 – mediated PIS, the authors indicate that Kay is involved but did not test Jra, the other part of the AP-1 complex, why not? The case would be stronger if Jra KD had the same effect as Kay KD.

3. Lines 210-11 : Interesting, free GFP proteins were also found in the hemolymph, which may be due to a cleavage of the C-terminus of Upd3 occurring after its secretion. Is UPD3 normally cleaved in the C terminus? What is the basis for this statement?

4. Fig. 3h - It is unclear how much hemolymph was loaded onto the gels. Were hemolymph extractions pooled or was the entire 0.5ul extracted from a single fly loaded into a single well? Data should be quantified and normalized to a loading control. What is the "Non-specific" band?

Minor Points-

1. For all figures, please clarify the genotypes of the "CNTRL" flies are (see also comment above).
2. Document contains typos and errors in English language usage that should be corrected.
2. Fig. 1 a - statistical tests used on these data should be a one-way ANOVA (3 genotypes).
3. Fig. 1 e,f&h - statistical tests used on these data should be a two-way ANOVA (2 genotypes and 2 conditions). Typos in legend for (f) and (g).
4. Fig. 2 a – It seems that the Venn diagram is from ref 13. This should be stated in the legend as well as ages of flies and some detail of the paraquat treatment from that study (how long, how much).
5. Fig. 2b – t-test is not appropriate; there are multiple genotypes and two treatments.
6. Fig. 2c – t-test not appropriate (3 genotypes and 2 treatments)
7. Fig. 2e – graph not consistent with others.
8. Fig. 2f – t-test not appropriate (3 genotypes and 2 treatments)
9. Fig. 2 I – what is the age of the flies being tested?
10. Fig. 3b – need to explain how "near cardiomyocyte nucleus" was determined.
11. Fig. 3e – t-test not appropriate (3 genotypes and 2 treatments)
12. Fig. 4h – what do the dashed lines represent?
13. Fig. 4i – one way ANOVA is needed
14. A control line has been included for Fig. 5 f & h but it is not clear to this reviewer what the genotype(s) is (are). If it is the oenocyte specific KD of Pex 5 by itself that is not a control per se (see Major point 1).

Reviewer #3 (Remarks to the Author):

The link between altered ROS homeostasis/peroxisomal function in hepatocyte-like oenocytes and cardiac aging is studied in a *Drosophila* model. In general, this is an extensive study that is well conducted with many knockdown/rescue setups. It is also appreciated that the authors sought for mechanistic insights underlying the increased expression of the inflammatory messenger Upd3 in oenocytes. The insights generated are novel and illustrate that peroxisome dysfunction in one tissue may affect distant tissues through the release of cytokines.

However, there are some discordances and shortcomings with regard to the primary cause of Upd3 upregulation in oenocytes and the 'exclusivity' of Upd3, which need to be resolved.

From Fig2b it is clear that knockdown of at least 8 different genes can decrease PQ induced arrhythmicity. In addition, Upd3 is secreted at high levels from other tissues. Still, oenocyte Upd3 is selected as the single cytokine mediating aging- and stress-induced cardiac arrhythmicity. Was an overexpression of Upd3 in other tissues at young ages performed?

The relation between the different paradigms in oenocytes causing arrhythmicity i.e. increased oxidative stress due to knockdown of catalase or SOD1 and the impaired import of peroxisomal proteins due to knockdown of Pex1, Pex5 is not clear. It is claimed that the knockdown of these Pex genes impacts on ROS metabolism but ROS levels were not determined. Importantly, it should be clarified why knockdown of Pex19 does not induce the same effects as compared to Pex1 and Pex5. This is incomprehensible as peroxisomal metabolism is obliterated with all Pex knockdowns. Furthermore, a potential involvement of altered lipid metabolism was disregarded but this was not thoroughly investigated. Dhapat was knocked down as a way to reduce plasmalogen levels but it was not assessed whether the levels of these lipids were effectively reduced in this time span. Phosphomevalonate kinase was suppressed but the relevance of this is obscure as the role of peroxisomes in cholesterol synthesis is still being debated (Wanders and Waterham, *Ann rev biochem*, 2006). Notably, the potential importance of α - and β -oxidation, both mentioned by the authors as essential functions of peroxisomes in the introduction and discussion, and of particular importance in tissues with high lipid metabolism such as oenocytes, was not addressed. This should also be assessed. This is essential because it was previously shown that suppression of the peroxisomal β -oxidation enzyme ACOX1, induces inflammatory genes including IL6 (El Hajj et al, *Endocrinology* 2012). This paper should be referred to.

The inclusion of the Pex1 G843D fibroblast line to validate the findings in the oenocytes is a strength of the manuscript. A drawback is that the analysis was limited to IL6 and P-JNK. At least, the consequences on ROS levels should also be included for a better correlation with the fly data. It should be clarified whether this is a homozygous mutant line.

PMP70 staining is used to identify peroxisomes. Although this is one of 3 ABC transporters that is often expressed in the peroxisomal membrane, this can not be considered as a marker for peroxisomes because its abundance in peroxisomes can vary according to the cell type. It is advised to identify peroxisomes using antibodies directed to PEX proteins, for example PEX14. Also, better use the terminology ABCD3 than PMP70.

Minor comments

Fig 5b: ICC of P-JNK is shown to be increased in Pex5 RNAi but it would be better to confirm this by western blotting

What is meant with the term peroxikinies??

In figure 8m a scheme is shown summarizing the data after overexpression of Pex5 in oenocytes, but it would be better to have a more general graph in which all data are compiled.

Throughout the manuscript there are many grammatical/typographical errors (plural (e.g. peroxisomes instead of peroxisome) – tenses – verbs)

Point-by-Point Response to Reviewers' Comments (#NCOMMS-19-16959):

We are grateful for reviewers' positive and constructive comments. Please find below our responses to each of the review comment. We have conducted suggested experiments and revised the manuscript accordingly. The source data underlying all figures are provided as a "Source Data" file. We hope that we have addressed all of the concerns and the revised manuscript has now met journal's publication criteria.

We have highlighted all major changes in red in the revised manuscript text, figures, and supplementary files. The minor changes (e.g. modifications in fly nomenclatures and figure labels) are not highlighted in red to avoid unnecessary confusion.

In the revised manuscript, we have conducted all suggested experiments. A total of 17 new results have now been added to the revised manuscript, as listed below.

1. Fig. 2j-k: SOHA analysis on flies overexpressing *upd3* in fat body/gut or in heart tissues.
2. Fig. 3c-d: QRT-PCR to measure cardiac *Socs36E* expression in flies with oenocyte-specific *upd3* KD under paraquat treatment or aging.
3. Fig. 3K: Quantification of the western blots measuring the circulation levels of *upd3*-GFP.
4. Fig. 4c: SOHA analysis on a new *Dhap-at* conditional knockout line.
5. Fig. 4c: SOHA analysis on oenocyte-specific knockdown of *ADPS*, another key enzyme involved in ether phospholipid biosynthesis.
6. Fig. 4c: SOHA analysis on oenocyte-specific knockdown of acyl-CoA oxidases (*Acox57D-d* and *Acox57D-p*), the key peroxisomal beta-oxidation enzymes.
7. Fig. 5a: QRT-PCR to measure the mRNA levels of *Jra* under *Pex5* KD.
8. Fig. 5d-e: Western blots to measure P-JNK from the dissected oenocytes with *Pex5* KD.
9. Fig. 5f: Measure the activity of a new JNK reporter (TRE-DsRed) in oenocytes with *Pex5* KD.
10. Fig. 5k: SOHA analysis on *Pex5 RNAi*; *Jra RNAi* double KD flies.
11. Fig. 7m: Modified model for peroxisome-mediated oenocyte-heart communication.
12. Supplementary Fig. 1c-e: Fluorescent imaging shows that both *PromE-Gal4* and *PromE-GeneSwitch-Gal4* drivers are specific to oenocytes, and no expression is found in cardiac tissues.
13. Supplementary Fig. 2c-e: Examine the effects of RU486 feeding on arrhythmia of three wild-type flies.
14. Supplementary Fig. 3d: Western blots to test the non-specific cross-activities of the anti-GFP antibody.
15. Supplementary Fig. 4a-c: Examine the STAT activity under paraquat treatment using *2XStat92E-GFP* and *10XStat92E-GFP* reporters.
16. Supplementary Fig. 5f-h: ROS measurements of flies with *Pex5*, *Pex1*, or *Pex19* KD.
17. Supplementary Fig. 5i: ROS measurements of human PEX1-G843D fibroblast cells.

Reviewers' comments:

Reviewer #1 (Remarks to the Author):

Huang and colleagues present compelling and interesting data exploring the effects of Oenocyte-mediated inflammation in the age-related decline of cardiac function in *Drosophila*. The authors show that increased oxidative stress in aging oenocytes results in the JNK/Kay-mediated expression of *Upd3* and subsequent activation of JAK/STAT signaling in cardiomyocytes, and that this non-autonomous signaling mechanism promotes cardiac arrhythmias. The authors further establish that the cause for the elevated ROS in aging oenocytes is the impairment of peroxisomal import, and, accordingly, show that factors that promote peroxisomal import in oenocytes can rescue age-related cardiac arrhythmias.

The work is expansive and logically presented, the data are overall of good quality and properly interpreted, and the manuscript is well - written and should be of interest to a broad audience. The topic addressed, namely how aging causes systemic inflammation and how dysfunction in specific tissues promotes dysfunction in others is of particular interest to the aging community and the presented data make a significant contribution to our understanding of the loss of homeostasis of the systemic milieu. It is likely that these findings can also be translated to vertebrate/human cases of cardiac dysfunction.

Author's response:

- We are grateful for reviewer's positive and constructive comments. We have conducted suggested experiments and revised the manuscript accordingly.

I have a few minor comments that the authors should address before the paper is published:

- The authors use STAT nuclear localization as a readout for STAT activity, but the images are not very clear, and it seems difficult to clearly establish nuclear localization of STAT in oenocytes. There are very good STAT reporters (10XSTAT-GFP and 2XSTAT-GFP) for flies, and the authors should use these to validate their results. Alternatively, one could also use qPCR to establish expression of *SOCS36E*, a target gene for STAT in flies.

Author's response:

- Thank you for the suggestions. We agree that the *Stat92E* nuclear localization is not clearly reflecting the STAT activity. We have performed suggested experiments to monitor STAT activity using either *Stat92E* reporters or *Socs36E* expression. We have successfully performed qRT-PCR to measure cardiac *Socs36E* expression and confirmed our previous observations using *Stat92E* immunostaining. Briefly, the cardiac expression of *Socs36E* was induced by paraquat treatment, while oenocyte-specific *upd3* KD diminished it (Fig. 3c). Age-dependent induction of *Socs36E* in the heart was also attenuated by oenocyte-specific *upd3* KD (Fig. 3d). We also tried two *Stat93E*-GFP reporters (2XStat92E-GFP and 10XStat92E-GFP). However, these reporters did not respond to paraquat treatment very well (Supplementary Fig. 4a-c). Thus, we did not further examine the effect of oenocyte-specific *upd3* KD using these reporters.

- Figs. 3 H and I are not informative and should be taken out of the manuscript. If STAT activation in cardiomyocytes by oenocyte-expressed *Upd3* can be clearly established (see above), then this experiment is unnecessary.

Author's response:

- We agree with reviewer's suggestions and have removed Fig. 3l. However, we decided to keep Fig. 3h (now Fig. 3j-k). The reason is that it has been long assumed that cytokines like upd3 are always secreted into circulation and can be easily detected in the hemolymph samples. However, this assumption has never been experimentally tested. Therefore, Fig. 3j-k are very important results and they provide directly evidence showing that upd3 is indeed released from oenocytes and can be detected in hemolymph.

- A critical control for many of the presented data is to show that the PromE and PromEGS drivers are indeed excluded from cardiomyocytes. The authors should show convincing evidence (images or western blots) showing that the drivers are only expressed in oenocytes, not the heart.

Author's response:

- Thank you for the suggestions. We have crossed *PromE-Gal4* and *PromE^{GS}-Gal4* with UAS-GFP reporters and showed that these drivers are specific to oenocytes. There is not expression detected in the heart (See Supplementary Fig. 1c-e).

Reviewer #2 (Remarks to the Author):

Summary: The submitted article, "Impaired peroxisomal import in *Drosophila* hepatocyte-like cells induces cardiac dysfunction through the pro-inflammatory cytokine Upd3" by Huang et al, uses the *Drosophila* genetic model to examine the possibility that liver-like oenocytes impact cardiac function in the fly. They convincingly show that aging induced cardiac arrhythmia and paraquat induced ROS stress can be blocked by oenocyte-specific over expression of SOD. They identify age-dependent increases in UPD3 expression in oenocytes and show that the effects on UPD3 are mediated by signaling through the JNK pathway. The authors show UPD3 from oenocytes is secreted into the hemolymph and is taken up by the heart. They also show that UPD3 induces JAK-STAT signaling in the heart. Consistent with their observations, oenocyte-specific KD of UPD3 prevented both the age-dependent increase in cardiac expression of STAT and cardiac arrhythmias. Their data supports the notion that it is a decrease in peroxisome import via PEX5 and not impaired mitochondrial function in oenocytes that is the cause of the increased UPD3 signaling. These findings are novel and extremely interesting as they suggest that non-cardiac peroxisomes play important non-autonomous roles in heart function via IL6/UPD3-mediated inter-organ communication.

The authors provide a significant amount of good data to support these conclusions. The data are overall compelling but there are some controls that are missing and see comments below.

Author's response:

- We are grateful for reviewer's positive and constructive comments. We have conducted suggested experiments, redone the statistical analyses, and revised the manuscript accordingly.

Major Points-

1. The RU-based Gene induction experiments are good but missing key controls, as feeding with RU by itself has some cardiac effects. The driver lines should be outcrossed to flies from the RNAi or OE genetic backgrounds and treated with and without RU to control for the effect of drug treatment and driver genetic background. Stock centers have the background insertion lines that should be used for these controls, (eg. the appropriate control for Fig. 5f would be a cross between the RNAi background line and the PromE GS line with the progeny being fed +/- RU).

Author's response:

- Thank you for the suggestions. We have examine the effects of RU486 feeding on arrhythmia by crossing *PromE^{GS}-Gal4* into three control lines, *attP40* RNAi (RNAi background line), *yw^R*, and *Gal4* RNAi. We found that RU486 feeding did not significantly affect cardiac arrhythmia (Supplementary Fig. 2c-e).

- Regarding Fig. 5f (now Fig. 5i), we were testing the genetic interaction between *Pex5* and *kay* in the regulation of *upd3* transcription. The background insertion lines (*attP40* RNAi) was indeed used as the control in this experiment (genotype information has been added to the figure legend). In this experiment, the control genotype is *UAS-Pex5^{RNAi}/attP40; PromE^{GS}/+* (fed with -RU or +RU), while the experimental genotype is *UAS-Pex5^{RNAi}/UAS-kay^{RNAi}; PromE^{GS}/+* (fed with -RU or +RU).

2. For the experiments to identify the mediators of *Pex5* – mediated PIS, the authors indicate that *Kay* is involved but did not test *Jra*, the other part of the AP-1 complex, why not? The case would be stronger if *Jra* KD had the same effect as *Kay* KD.

Author's response:

- Thank you for the suggestions. We have now included experiments testing the role of *Jra*. Similar to *kay*, *Jra* was up-regulated by *Pex5* KD (Fig. 5a), and *Jra* KD also attenuated *Pex5* RNAi-induced arrhythmia (Fig. 5k).

3. Lines 210-11 : Interesting, free GFP proteins were also found in the hemolymph, which may be due to a cleavage of the C-terminus of *Upd3* occurring after its secretion. Is *UPD3* normally cleaved in the C terminus? What is the basis for this statement?

Author's response:

- Although the proteolytic processing of *upd3* is not fully understood, it is known that the activities of many mammalian cytokines are regulated by proteolytic processing (Fu et al., 2017), and *IL-6* is known to be cleaved by meprin metalloproteases at its c-terminus (Keiffer et al., 2014). Future studies are needed to carefully examine the proteolytic processing of fly *upd3*.

4. Fig. 3h - It is unclear how much hemolymph was loaded onto the gels. Were hemolymph extractions pooled or was the entire 0.5ul extracted from a single fly loaded into a single well? Data should be quantified and normalized to a loading control. What is the "Non-specific" band?

Author's response:

- Sorry for the confusion. We have revised the method section accordingly. Briefly, about 0.5 µl of hemolymph extracted from 30 flies was pooled in a 1.5 ml microcentrifuge tube

containing 10 µl of 1x PBS and protein inhibitor cocktail. Then equal amount of hemolymph (0.5 µl in 10 µl of 1x PBS) was denature and loaded on SDS-PAGE gels.

- We have quantified and normalized the western blot data to total protein (Fig. 3k). We tried to obtain an anti-Lsp2 antibody to be used as the loading control for hemolymph samples, but without any success. Therefore, total proteins were used in normalization.

- The non-specific band (~70kDa) was resulted from the non-specific reaction between the anti-GFP antibody and fly protein extracts. We found similar non-specific bands in several other western blots using anti-GFP against protein extracts from wild-type flies.

Minor Points-

1. For all figures, please clarify the genotypes of the “CNTRL” flies are (see also comment above).

Author’s response:

- Thank you for your suggestion, we have included control genotype information in each figure legend.

2. Document contains typos and errors in English language usage that should be corrected.

Author’s response:

- We have reviewed the manuscript and fixed the typos and grammar errors accordingly.

2. Fig. 1 a - statistical tests used on these data should be a one-way ANOVA (3 genotypes).

Author’s response:

- We have redone the analysis with correct statistical tests.

3. Fig. 1 e,f&h - statistical tests used on these data should be a two-way ANOVA (2 genotypes and 2 conditions). Typos in legend for (f) and (g).

Author’s response:

-We have redone the analysis with correct statistical tests

-Typos have been corrected.

4. Fig. 2 a – It seems that the Venn diagram is from ref 13. This should be stated in the legend as well as ages of flies and some detail of the paraquat treatment from that study (how long, how much).

Author’s response:

-We have clarified this by adding the Ref 13 to the figure legend.

-We have included details on fly ages and PQ treatment (dosage and duration) in the figure legend.

5. Fig. 2b – t-test is not appropriate; there are multiple genotypes and two treatments.

Author’s response:

We have redone the analysis with correct statistical tests.

6. Fig. 2c – t-test not appropriate (3 genotypes and 2 treatments)

Author's response:

We have redone the analysis with correct statistical tests.

7. Fig. 2e – graph not consistent with others.

Author's response:

-We have modified the graph to make it consistent with other graphs.

8. Fig. 2f – t-test not appropriate (3 genotypes and 2 treatments)

Author's response:

We have redone the analysis with correct statistical tests.

9. Fig. 2 I – what is the age of the flies being tested?

Author's response:

- Fly age information has been added to the figure legend. Flies were 1-week-old.

10. Fig. 3b – need to explain how “near cardiomyocyte nucleus” was determined.

Author's response:

- Detailed description has been include in the method section. See below,
To quantify the punctae near the nucleus, we first selected ROIs surround the nucleus according to Hoechst signal, and then counted the punctae number within each ROI using the CellSens “Measure and Count” module.

11. Fig. 3e – t-test not appropriate (3 genotypes and 2 treatments)

Author's response:

- We have redone the analysis with correct statistical tests.

12. Fig. 4h – what do the dashed lines represent?

Author's response:

- Dashed lines have been removed. It was representing the boundary between heart and pericardial cells.

13. Fig. 4i – one way ANOVA is needed

Author's response:

- We have redone the analysis with correct statistical tests.

14. A control line has been included for Fig. 5 f & h but it is not clear to this reviewer what the genotype(s) is (are). If it is the oenocyte specific KD of Pex 5 by itself that is not a control per se (see Major point 1).

Author's response:

- Genotype information added to the figure legend. The control genotype here is *UAS-Pex5^{RNAi}/attP40; PromE^{GS}/+*.

Reviewer #3 (Remarks to the Author):

The link between altered ROS homeostasis/peroxisomal function in hepatocyte-like oenocytes and cardiac aging is studied in a *Drosophila* model. In general, this is an extensive study that is well conducted with many knockdown/rescue setups. It is also appreciated that the authors sought for mechanistic insights underlying the increased expression of the inflammatory messenger Upd3 in oenocytes. The insights generated are novel and illustrate that peroxisome dysfunction in one tissue may affect distant tissues through the release of cytokines.

Author's response:

- We are grateful for reviewer's positive and constructive comments. We have conducted suggested experiments and revised the manuscript accordingly.

However, there are some discordances and shortcomings with regard to the primary cause of Upd3 upregulation in oenocytes and the 'exclusivity' of Upd3, which need to be resolved.

From Fig2b it is clear that knockdown of at least 8 different genes can decrease PQ induced arrhythmicity. In addition, Upd3 is secreted at high levels from other tissues. Still, oenocyte Upd3 is selected as the single cytokine mediating aging- and stress-induced cardiac arrhythmicity. Was an overexpression of Upd3 in other tissues at young ages performed?

Author's response:

- Thank you for the suggestions. We have performed suggested experiments to overexpress upd3 in both heart tissue (*Hand>Upd3^{OE}*) and fat body/gut (*S106^{GS}>Upd3^{OE}*). Interestingly, expressing upd3 in these tissues did not elevate arrhythmia (Fig. 2j-k).

The relation between the different paradigms in oenocytes causing arrhythmicity i.e. increased oxidative stress due to knockdown of catalase or SOD1 and the impaired import of peroxisomal proteins due to knockdown of Pex1, Pex5 is not clear. It is claimed that the knockdown of these Pex genes impacts on ROS metabolism but ROS levels were not determined. Importantly, it should be clarified why knockdown of Pex19 does not induce the same effects as compared to Pex1 and Pex5. This is incomprehensible as peroxisomal metabolism is obliterated with all Pex knockdowns. Furthermore, a potential involvement of altered lipid metabolism was disregarded but this was not thoroughly investigated. Dhapat was knocked down as a way to reduce plasmalogen levels but it was not assessed whether the levels of these lipids were effectively reduced in this time span. Phosphomevalonate kinase was suppressed but the relevance of this is obscure as the role of peroxisomes in cholesterol synthesis is still being debated (Wanders and Waterham, Ann rev biochem, 2006). Notably, the potential importance of α - and β -oxidation, both mentioned by the authors as essential functions of peroxisomes in the introduction and discussion, and of particular importance in tissues with high lipid metabolism such as oenocytes, was not addressed. This should also be assessed. This is essential because it was previously shown that suppression of the peroxisomal β -

oxidation enzyme ACOX1, induces inflammatory genes including IL6 (El Hajj et al, Endocrinology 2012). This paper should be referred to.

Author's response:

- Thank you for the suggestions. We have measured oenocyte ROS levels in *Pex1* KD, *Pex5* KD, or *Pex19* KD. We found that knocking down both *Pex1* and *Pex5* induced ROS in oenocytes (Supplementary Fig. 5f-g). Interestingly, knockdown of *Pex19* did not elevate ROS (Supplementary Fig. 5h). The different regulation on ROS metabolism by different peroxines might explain their distinct roles on cardiac arrhythmia and oenocyte-heart communication.

- Regarding plasmalogen measurement, we tried to detect it using Folch extraction and LC-MS/MS in both S2 and adult flies. However, the levels of plasmalogen are very low and it is hard to obtain reliable quantitative data with our current method. Until we optimize our detection method, we cannot exclude the potential effect of plasmalogen in oenocyte-heart communication.

- On the other hand, we did carefully examine the plasmalogen biosynthesis pathway using a new Dhap-at conditional knockout line, as well as RNAi against ADPS (another key enzyme in plasmalogen biosynthesis). Both genetic manipulations in oenocytes did not affect cardiac arrhythmia (Fig. 4c).

- Due to the debatable role of peroxisomes in cholesterol synthesis, we have removed the data related to phosphomevalonate kinase (CG10268).

- To test the potential involvement of ACOX1, we knocked down two predicted ACOX1 orthologues (*Acox57D-d* and *Acox57D-p*) in fly oenocytes. Again, we did not observe an induction of arrhythmia. It is likely that peroxisomal ROS homeostasis plays a major role in mediating oenocyte-heart communication.

- Although El Hajj et al., Endocrinology 2012 reported an increased level of IL6 in ACOX1 deficiency fibroblasts from P-NALD patients, it is not known whether fibroblast-produced IL6 impacts cardiac function. Based on our tissue-specific *upd3* overexpression experiments (Fig. 2j-k), we now know that *upd3* produced from different tissues exhibits distinct effects on cardiac arrhythmicity. We have included this citation and a discussion in the revision.

The inclusion of the *Pex1* G843D fibroblast line to validate the findings in the oenocytes is a strength of the manuscript. A drawback is that the analysis was limited to IL6 and P-JNK. At least, the consequences on ROS levels should also be included for a better correlation with the fly data. It should be clarified whether this is a homozygous mutant line.

Author's response:

Thank you for the suggestions.

- We have measured ROS levels in PEX1-G843D cell line. It also shows elevated levels of H₂O₂ (Supplementary Fig. 5i).

- The cell line is a homozygous mutant line. We have added this information to the method section.

PMP70 staining is used to identify peroxisomes. Although this is one of 3 ABC transporters that is often expressed in the peroxisomal membrane, this can not be considered as a marker for peroxisomes because its abundance in peroxisomes can vary according to the cell type. It is advised to identify peroxisomes using antibodies directed to PEX proteins, for example PEX14. Also, better use the terminology ABCD3 than PMP70.

Author's response:

- Thank you for the suggestions. We realized the limitation of using PMP70 antibodies. It may not label all peroxisomes in the cells. We have tried to generate fly Pex14 polyclonal antibodies, however, we encountered several issues, such as high background and low specificity. Unfortunately, we could not find and produce effective antibody against fly Pex14 to address reviewer's comments.
- To be consistent with fly nomenclature, Pmp70 is used, instead of ABCD3.

Minor comments

Fig 5b: ICC of P-JNK is shown to be increased in Pex5 RNAi but it would be better to confirm this by western blotting

Author's response:

- Thank you for the suggestions. We have performed western blots on dissected oenocytes to examine the phosphorylation of JNK. Consistently, we found that *Pex5* KD slightly increased the phosphorylation of JNK (Fig. 5d-e). In addition, we also utilized a JNK reporter (TRE-DsRedT4) to monitor the transcription activity of AP-1 complex. *Pex5* KD significantly induced the reporter activity in oenocytes (Fig. 5f).

What is meant with the term peroxikinies??

Author's response:

- Sorry, it is a typo. It should be "peroxikines". We have corrected it in the revision.

In figure 8m a scheme is shown summarizing the data after overexpression of Pex5 in oenocytes, but it would be better to have a more general graph in which all data are compiled.

Author's response:

- Thank you for the suggestions. We have replaced it with a new diagram summarizing all the findings (Fig. 7m).

Throughout the manuscript there are many grammatical/typographical errors (plural (e.g. peroxisomes instead of peroxisome) – tenses – verbs)

Author's response:

- We have reviewed the manuscript and corrected the typos and grammar errors accordingly.

Reviewers' comments:

Reviewer #1 (Remarks to the Author):

The authors have responded appropriately to my comments and have added new relevant data that strengthen the manuscript. I believe the manuscript can now be recommended for publication.

Reviewer #2 (Remarks to the Author):

The authors have provided a significant amount of new data and explanations to their manuscript. I feel that the authors have adequately responded to the reviewers' comments. It is still advisable to have the grammar/ word usage checked by a native English speaker as there are problems at a number of spots in the text. Nevertheless, their findings are novel and extremely interesting, suggesting that non-cardiac peroxisomes play important non-autonomous roles in heart function via IL6/UPD3-mediated inter-organ communication, and deserving of publication in Nature Communications.

Reviewer #3 (Remarks to the Author):

Several issues were well addressed, including a better analysis of the potential involvement of peroxisomal metabolic pathways in the observed effects.

My question why Pex19 knockdown deviates from Pex1, Pex14 and Pex5 was not clarified satisfactorily. A potential underlying cause may be that the (partial) knockdown did not result in a defect in peroxisomal matrix import. In fact, the authors validate the knockdown of Pex1 and Pex5 by performing qPCR but do not show data on Pex19. Furthermore, it is possible that even low levels of residual PEX19 may support peroxisome biogenesis. Therefore, it is necessary to assess the knockdowns by a functional assay and show whether or not import competent peroxisomes have been deleted.

Secondly, I requested to measure ROS levels in the KD oenocytes. These experiments were performed but the results are not convincing. The Pex1 and Pex5 knockdown seem to result in increased DHE levels. My concern is that it is obvious that in the –RU condition for the Pex19 KD, the DHE staining is more intense than in other –RU controls. Why is there a difference? Furthermore,

this is not reflected in the % DHE values in the graphs. This raises doubts on the reliability of the DHE data. Because DHE stains nuclei, it would be better to normalize the DHE to DAPI fluorescence.

Upon my request, the authors measured ROS levels in Pex1G843D patient fibroblasts. In the text they claim that ROS is increased but from the figure it is clear that this does not reach statistical significance due to large variability. It is also strange that here another assay to assess ROS levels is used (Amplex Red). In the methods it is mentioned that 'different cell lines' were analyzed and that the experiments were performed in triplicate. It is not clear how many independent lines were used.

The authors launch the term peroxikine but it should be better determined what this stands for.

I am sorry that I did not comment on the final sentence in the discussion in my first review. However, I want to point out that this is highly speculative. At this point there is no indication that the effects of peroxisome deletion in oenocytes/liver on heart function can be translated from the fly to human 'aging diseases'.

Point-by-Point Response to Reviewers' Comments (#NCOMMS-19-16959):

We are grateful for reviewers' positive and constructive comments. Please find below our responses to each of the review comment. We have conducted all suggested experiments and revised the manuscript accordingly. The source data underlying all figures are provided as a "Source Data" file. We hope that we have addressed all of the concerns and the revised manuscript has now met journal's publication criteria.

We have highlighted all major changes in red in the revised manuscript text, figures, and supplementary files. A total of 6 new results have been added to the revised manuscript, as listed below.

1. Supplementary Fig. S5f and g, QRT-PCR analysis to validate the knockdown of *Pex19* and *Pex14* genes.
2. Fig 6i-l, examining the effects of *Pex19* knockdown on peroxisomal import function.
3. Fig. 6k, examining the effects of *Pex19* knockdown on peroxisome biogenesis.
4. Fig. 4d, examining the effects of *Pex19* knockdown on *upd3* induction in fly oenocytes.
5. Supplementary Fig. S5h-j, new DHE quantification by normalizing the DHE signals to DAPI.
6. Supplementary Fig. S5k, ROS measurements of human PEX1-G843D fibroblast cells (6 biological replicates included).

Reviewers' comments:

Reviewer #1 (Remarks to the Author):

The authors have responded appropriately to my comments and have added new relevant data that strengthen the manuscript. I believe the manuscript can now be recommended for publication.

Author's response:

- We are grateful for reviewer's positive comments and helps throughout the peer-review process.

Reviewer #2 (Remarks to the Author):

The authors have provided a significant amount of new data and explanations to their manuscript. I feel that the authors have adequately responded to the reviewers' comments. It is still advisable to have the grammar/ word usage checked by a native English speaker as there are problems at a number of spots in the text. Nevertheless, their findings are novel and extremely interesting, suggesting that non-cardiac peroxisomes play important non-autonomous roles in heart function via IL6/UPD3-mediated inter-organ communication, and deserving of publication in Nature Communications.

Author's response:

- We are grateful for reviewer's positive comments and helps throughout the peer-review process. We have carefully checked the grammar errors and typos, and revised the manuscript accordingly.

Reviewer #3 (Remarks to the Author):

Several issues were well addressed, including a better analysis of the potential involvement of peroxisomal metabolic pathways in the observed effects. My question why *Pex19* knockdown deviates from *Pex1*, *Pex14* and *Pex5* was not clarified satisfactorily. A potential underlying cause may be that the (partial) knockdown did not result in a defect in peroxisomal matrix import. In fact, the authors validate the knockdown of *Pex1* and *Pex5* by performing qPCR but do not show data on *Pex19*. Furthermore, it is possible that even low levels of residual PEX19 may support peroxisome biogenesis. Therefore, it is necessary to assess the knockdowns by a functional assay and show whether or not import competent peroxisomes have been deleted.

Author's response:

- Thank you for reviewer's thoughtfulness and constructive comments. We agree all reviewer's suggestions. To address the concerns, we have validated the knockdown (KD) of *Pex19*, as well as *Pex14* (see Supplementary Fig. S5f and g). The qPCR results showed a significant knockdown of both *Pex19* and *Pex14* compared to control.

We also examined the effects of *Pex19* knockdown on peroxisomal import function (Fig 6i-l). Unlike *Pex15* KD (Fig. 6h), *Pex19* KD did not alter peroxisomal protein import (indicated by normalized SKL-positive punctae), which may explain why *Pex19* KD did not impact heart function, ROS levels, and the induction of *upd3* (new results, see Fig. 4d). Although *Pex19* RNAi resulted in a partial reduction of its expression (about 70%), the number of peroxisomes (indicated by PMP70-positive punctae number) in *Pex19* KD flies was significantly reduced (Fig. 6k), suggesting that *Pex19* KD impaired peroxisome biogenesis but not import function.

Secondly, I requested to measure ROS levels in the KD oenocytes. These experiments were performed but the results are not convincing. The *Pex1* and *Pex5* knockdown seem to result in increased DHE levels. My concern is that it is obvious that in the -RU condition for the *Pex19* KD, the DHE staining is more intense than in other -RU controls. Why is there a difference? Furthermore, this is not reflected in the % DHE values in the graphs. This raises doubts on the reliability of the DHE data. Because DHE stains nuclei, it would be better to normalize the DHE to DAPI fluorescence.

Author's response:

- Thank you for reviewer's comments. The difference in DHE levels between *Pex19* KD and other -RU controls is likely due to genetic background variations between these RNAi lines. See below for detailed genetic backgrounds:

- *Pex19* RNAi: $y[1] sc[*] v[1] sev[21]; P\{y[+7.7] v[+1.8]=TRiP.HMC03104\}attP2;$
- *Pex1* RNAi: $y[1] v[1]; P\{y[+7.7] v[+1.8]=TRiP.HM05190\}attP2;$
- *Pex5* RNAi: $y[1] v[1]; P\{y[+7.7] v[+1.8]=TRiP.HMJ21920\}attP40.$

It is clear that the X chromosome of *Pex19* RNAi ($y[1] sc[*] v[1] sev[21]$) is different from *Pex1* and *Pex5* RNAi ($y[1] v[1]$). Fortunately, the GeneSwitch system used in the present study eliminates the genetic background issues. For each knockdown experiment, all

flies have the same genetic background. The gene knockdown was induced by RU486 (mifepristone) feeding.

As suggested by the reviewer, we have redone the DHE quantification by normalizing the DHE signals to DAPI (Supplementary Fig. S5h-j). The new quantification results agree with our previous conclusion.

Upon my request, the authors measured ROS levels in Pex1G843D patient fibroblasts. In the text they claim that ROS is increased but from the figure it is clear that this does not reach statistical significance due to large variability. It is also strange that here another assay to assess ROS levels is used (Amplex Red). In the methods it is mentioned that 'different cell lines' were analyzed and that the experiments were performed in triplicate. It is not clear how many independent lines were used.

Author's response:

- We have now included 6 biological replicates of the ROS measurement (Supplementary Fig. S5k). Although there is a large variability among the replicates, the unpaired student's t-test (two-tailed) showed a significant difference between the wild-type and Pex1G843D fibroblasts ($p=0.043$).

In addition, the reason to use Amplex Red, instead of DHE to measure ROS is because the elevated ROS in mammalian peroxin mutants (measured by DCF-DA) has been reported previously (Piao et al., *Antioxidants and Redox Signaling*, 2018). Here, we were specifically looking at the levels of hydrogen peroxide (H₂O₂) and peroxidase activity in Pex1G843D fibroblasts. Amplex Red method has been widely used for H₂O₂ measurement in many previous studies. We have included a sentence in the result section to clarify the use of Amplex Red.

Lastly, we apologize about the typo on 'different cell lines'. There is only one Pex1G843D fibroblasts line used in the present study. We have corrected this statement accordingly.

The authors launch the term peroxikine but it should be better determined what this stands for.

I am sorry that I did not comment on the final sentence in the discussion in my first review. However, I want to point out that this is highly speculative. At this point there is no indication that the effects of peroxisome deletion in oenocytes/liver on heart function can be translated from the fly to human 'aging diseases'.

Author's response:

- Similar to 'mitokine, we define 'peroxikines' as hormonal factors that are produced and released in response to peroxisomal stresses (such as impaired peroxisomal import) to modulate cellular homeostasis in distant tissues. We have expanded the description of "peroxikine" in both results and discussion sections.

Regarding the last sentence in our discussion, we agree that our statement is speculative. We have revised the last sentences to the following: "Our findings suggest that peroxisome is a vital organelle and central regulator of inflammaging and inter-tissue communication. Future work will be of interests to examine the role of liver peroxisomes in age-related cardiac diseases in mammalian systems."

REVIEWERS' COMMENTS:

Reviewer #3 (Remarks to the Author):

By performing additional validation experiments of the Pex19 knockdown, which I suggested, the anomalies with the Pex5/Pex14 depleted cells could now be clarified.

The other issues were also satisfactorily dealt with by the authors.